# Evaluation of therapeutic PD-1 antibodies by an advanced single-molecule imaging system detecting human PD-1 microclusters

Wataru Nishi [1,2], Ei Wakamatsu [2], Hiroaki Machiyama [2], Ryohei Matsushima[1,2], Kensho Saito[2,3], Yosuke Yoshida[2,4], Tetsushi Nishikawa[2,5], Tomohiro Takehara [6], Hiroko Toyota[2], Masae Furuhata[2], Hitoshi Nishijima[2], Arata Takeuchi[2], Miyuki Azuma [7], Makoto Suzuki[1] & Tadashi Yokosuka [2] ✉

With recent advances in immune checkpoint inhibitors (ICIs), immunotherapy has become the standard treatment for various malignant tumors. Their indications and dosages have been determined empirically, taking individually conducted clinical trials into consideration, but without a standard method to evaluate them. Here we establish an advanced imaging system to visualize human PD-1 microclusters, in which a minimal T cell receptor (TCR) signaling unit co-localizes with the inhibitory co-receptor PD-1 in vitro. In these microclusters PD-1 dephosphorylates both the TCR/CD3 complex and its downstream signaling molecules via the recruitment of a phosphatase, SHP2, upon stimulation with the ligand hPD-L1. In this system, blocking antibodies for hPD-1-hPD-L1 binding inhibits hPD-1 microcluster formation, and each therapeutic antibody (pembrolizumab, nivolumab, durvalumab and atezolizumab) is characterized by a proprietary optimal concentration and combinatorial efficiency enhancement. We propose that our imaging system could digitally evaluate PD-1-mediated T cell suppression to evaluate their clinical usefulness and to develop the most suitable combinations among ICIs or between ICIs and conventional cancer treatments.

Among various costimulatory and coinhibitory receptors, the latter ones specifically expressed on T cells are called immune checkpoint molecules and include cytotoxic T lymphocyte antigen-4 (CTLA-4), programmed cell death-1 (PD-1), lymphocyte activation gene 3 (LAG-3), T cell immunoreceptor with Ig and ITIM domains (TIGIT), T cell immunoglobulin mucin 3 (Tim-3), and so on[1]. Immune checkpoint inhibitor (ICI) therapy is becoming a standard treatment for various types of malignant tumors. ICIs generally block the binding of these checkpoint molecules to their ligands and are being actively studied to elucidate not only their epidemiology but also their molecular mechanisms to suppress tumor growth by rescuing from the exhausted status of T cells in tumors[2-6].

PD-1, which is only expressed 2–6 h after T cell receptor (TCR) stimulation of naive T cells and is kept on effector cells for more than a week, contributes to T cell suppression at all stages from priming to effector and memory differentiation[7]. PD-1 possesses two ligands, PD-

[1]Department of Thoracic Surgery, Graduate School of Medical Sciences, Kumamoto University, Kumamoto 860-8556, Japan. [2]Department of Immunology, Tokyo Medical University, Tokyo 160-8402, Japan. [3]School of Life Sciences, Tokyo University of Pharmacy and Life Sciences, Tokyo 192-0392, Japan. [4]Department of Nephrology, Tokyo Medical University, Tokyo 160-8402, Japan. [5]Department of Dermatology, Tokyo Medical University, Tokyo 160-0023, Japan. [6]Division of Pulmonary Medicine, Department of Medicine, Keio University School of Medicine, Tokyo 160-8582, Japan. [7]Department of Molecular Immunology, Graduate School of Medical and Dental Sciences, Tokyo Medical and Dental University, Tokyo 113-8549, Japan. ✉e-mail: yokosuka@tokyo-med.ac.jp

1-ligand 1 (PD-L1) and PD-L2; the former is ubiquitously and continuously expressed particularly on vascular endothelial and epithelial cells as well as lymphoid cells to maintain adequate immune responses[8–10] and the latter is preferentially expressed on immune cells such as professional antigen-presenting cells (APC)[8,11,12]. Recent clinical reports have described malignant tumors with high expression of human (h) PD-L2 or with dual expression of hPD-L1 and hPD-L2 as a ligand for hPD-1[13]. In our previous report, we confirmed the endogenous expression of hPD-L1 and hPD-L2 by different cell lines of human lung cancers before or after in vitro culture with interferon γ (IFNγ) and found various patterns of hPD-1 ligand expression depending on each line[14]. Many types of ICIs have been developed and approved for therapies of these types of cancers all over the world[15–17]. Each company has added their own features to each drug and have determined both indications and dosages based on the results of clinical trials separately conducted for each drug. We have minimal information about the head-to-head functional comparison among the ICIs and the optimum effective dose for the body weight and age of each patient[18–20]. Several combination therapies of ICIs with other ICIs or conventional cancer treatments have recently been introduced and have shown some efficacy[3,21]. As such, the indications for ICIs are becoming increasingly expanded. This may benefit patients and save medical care costs by reducing the resources needed to determine appropriate ICI dosing and combination therapies.

Recent imaging technologies unveiled the distinct structure for antigen recognition and signaling pathways via TCRs at the adhesion interface between a T cell and an APC, which we called an immunological synapse[22,23], and which was further recognized as the minimal signaling unit called a TCR microcluster signalosome[24–26]. In our previous report, using a supported lipid bilayer (SLB) expressing MHC molecules and murine (m) PD-L1, we described that TCR microclusters are colocalized by PD-1 after PD-1-PD-L1 binding. PD-1 could recruit the phosphatase, SHP2, at the TCR microclusters and attenuate the dephosphorylation status of a TCR/CD3 complex, CD28, and almost all of the signaling molecules downstream of TCRs through forming PD-1-mediated inhibitory microclusters[14,27]. We also demonstrated the induction of PD-1 microcluster formation triggered by PD-1-PD-L2 binding, the predominant binding capacity of PD-L2 comparing PD-L1 with a higher affinity of PD-L2 toward PD-1, and the physical competition between PD-L1 and PD-L2 in PD-1 binding.

Most of the affinities between receptors and ligands are generally measured by the basic strategy, surface plasmon resonance (SPR) spectrometry, which examines the 1:1 molecular interaction between a membrane-attached receptor and a soluble ligand. However, we sometimes observed obvious differences in the theoretical data from SPR spectrometry, which usually ignore the membrane–membrane interaction and the lateral or cis binding of receptors or ligands. The SLB is an innovative method that can evaluate the affinities under preferable conditions that allow the lateral migration of both receptors and ligands expressed on a cell membrane.

Here, we construct a new super-resolution imaging system to visualize hPD-1 binding to hPD-L1 and/or hPD-L2 and observe the microcluster formation by hPD-1. Because the entire system is completely adapted to human materials, we can evaluate the efficacies of anti-PD-1, PD-L1, or PD-L2, such as nivolumab, pembrolizumab, atezolizumab, or durvalumab, which are used in clinical practice. We find that hPD-1 forms clusters triggered by hPD-L1 or hPD-L2 binding at a T cell–APC or –SLB interface generating inhibitory microclusters, which dephosphorylate both TCR/CD3 complexes and their downstream signaling molecules via recruitment of the phosphatase, SHP2, in a similar fashion, as does mPD-1. Furthermore, each antibody requires its own concentration to block the PD-1–PD-L1 or –PD-L2 binding, and acquires much more effective functions in these blockings through the combination of other antibodies against the different

target molecules, but not of the same one. We believe that our newly established molecular imaging system for hPD-1 will allow us to evaluate the T cell activation and practical functions of each ICI digitally by examining the correlation between such biological responses and clearly visualized microcluster formation. As a variety of new ICIs are being researched, this study may have a wide range of applications to estimate the blocking effects, to calculate the real concentrations required as effective ICIs, and to select the combination of these ICIs and different treatments, such as other ICIs and conventional cancer therapies.

## Results

### hPD-L1 induces clustering of hPD-1 at TCR microclusters

We attempted to apply our previous report to analyze the localization of hPD-1 more accurately by using the new imaging system constructed by a high-resolution total internal reflection fluorescence (TIRF) microscope and SLBs consisting of glycosylphosphatidylinositol-anchored murine major histocompatibility complex Class II (MHC-II) molecule, I-E$^k$, (I-E$^k$-GPI) and mICAM-1 as basic components, to which hPD-L1 can be added as a ligand for hPD-1. To determine the density of hPD-L1 on SLBs suitable for imaging hPD-1 microclusters, we examined the clustering of hPD-1 at various hPD-L1 densities. We confirmed that those between 37.5 and 150 molecules/m² were appropriate and that the cell surface densities of hPD-L1 in several human cancer cell lines were within that range (Supplementary Fig. 1a−c). Based on both these experiments and several previous reports visualizing PD-1 microclusters[14,27], the number of hPD-L1 molecules on an SLB was determined to be 150 molecules/μm², which could be within the physiological range. Splenic CD4$^+$ T cells were prepared from AND-TCR, whose specificity for moth cytochrome c 88–103 (MCC$_{88–103}$) on I-E$^k$, transgenic (Tg) Rag2- (Rag2$^{-/-}$) and PD-1-deficient (Pdcd1$^{-/-}$) mice, stimulated, and retrovirally transduced by enhanced green fluorescent protein-tagged hPD-1 (hPD-1-EGFP) (Supplementary Fig. 1d). These T cells were plated onto SLBs in the above conditions and imaged by confocal or TIRF microscopy. In the presence of hPD-L1 on the SLB, hPD-1 formed clusters at the nascent T cell-bilayer contact regions and the hPD-1 microclusters migrated toward the center of an immunological synapse, forming central-supramolecular activation clusters (c-SMAC) (Fig. 1a, b and Supplementary Movie 1). To examine the colocalization of hPD-1 and TCRs within the microclusters, these AND-Tg CD4$^+$ T cells expressing hPD-1-EGFP were prestained with DyLight 650-labeled anti-TCRβ (H57) Fab and imaged on the same SLB as in Fig. 1a. In the presence of hPD-L1, hPD-1 were initially accumulated at the same cluster together with TCRs (hPD-1-TCR microclusters[28,29]) (Fig. 1c, left, and 1d), and then these hPD-1 clusters eventually translocated toward the c-SMAC (Fig. 1c, right). Furthermore, hPD-1-EGFP reconstituted into AND-TCR-expressing T cell hybridoma (2D12) deficient in mPD-1 by CRISPR-Cas9 showed similar behaviors as hPD-1 in primary T cells (Supplementary Fig. 1e−g). To examine the colocalization of hPD-1 and hPD-L1 at a T cell−APC interface, we conjugated the T cells expressing hPD-1-EGFP with an I-E$^{k+}$ APC line, DC-1 cells, introduced with hPD-L1 tagged by HaloTag (Supplementary Fig. 1h), and confirmed the accumulation of hPD-1 and hPD-L1 at a T cell−APC interface only if hPD-L1 was expressed (Fig. 1e). These data clearly demonstrated that hPD-1 formed clusters together with TCR at the same region dependently on the binding to its ligand, hPD-L1, in the same manner shown by mPD-1.

### SHP2 transiently translocates at hPD-1 microclusters triggered by hPD-1-hPD-L1 binding to suppress T cell activation

We next examined whether hPD-1 recruit the phosphatase, SHP2, toward TCR microclusters to form an inhibitory signalosome in the same way as mPD-1. AND-TCR T cells co-expressing hPD-1-HaloTag and EGFP-SHP1 or -SHP2 were settled on the SLBs with hPD-L1, demonstrating that hPD-1 microclusters colocalized with SHP2, but not with SHP1 (Fig. 2a). These results were confirmed by cell-cell

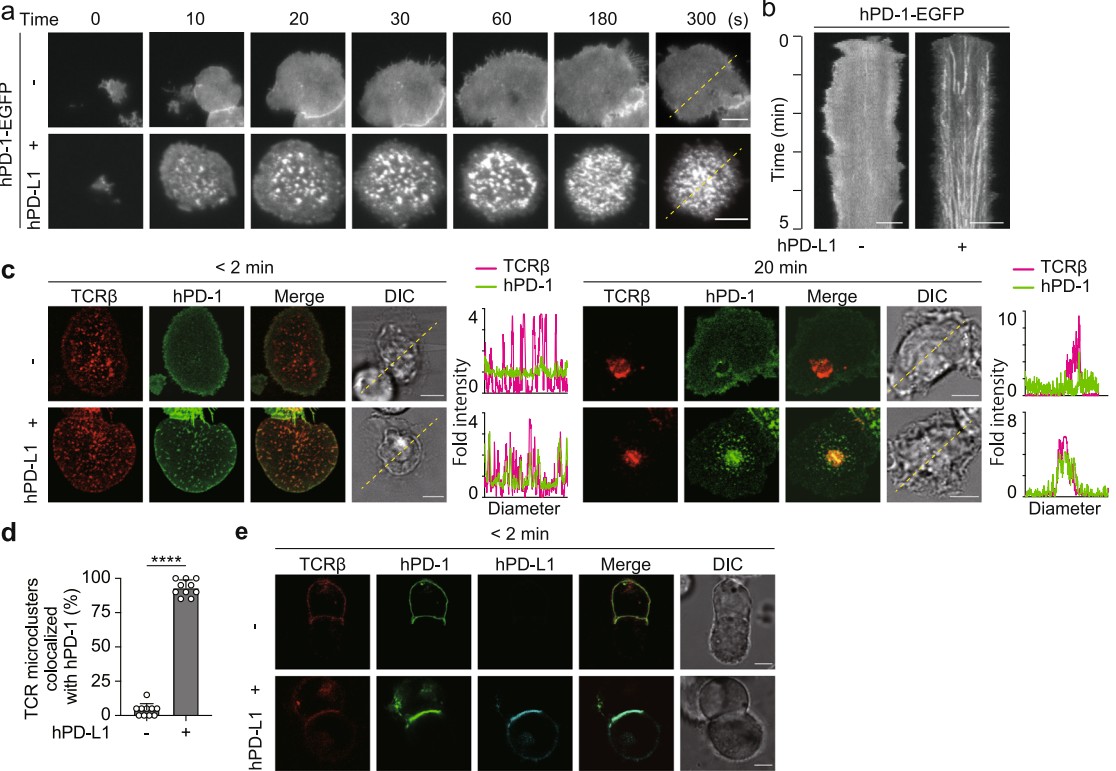

**Fig. 1 | hPD-1 gathers with TCR to form microclusters at a T cell-bilayer interface in a ligand-binding fashion. a** AND-TCR-Tg CD4+ T cells expressing hPD-1–EGFP (green) were plated onto an MCC$_{88-103}$-prepulsed SLB containing I-E$^k$–, mICAM-1–, and without (top) or with hPD-L1–GPI (bottom) and real-time imaged by TIRF microscopy (times are above images; Supplementary Movie 1). **b** Clustering and centripetal movement of hPD-1 on the diagonal yellow lines in **a** are presented as horizontal elements in kymographs. **c** The cells in **a** were prestained with DyLight 650-labeled anti-TCRβ (H57) Fab (red), plated onto an SLB as in **a**, and real-time imaged by confocal microscopy 2 (left) or 20 min (right) after contact. Histograms show fold fluorescent intensities of TCRβ (magenta) and hPD-1 (green) on the

diagonal yellow lines in the DIC images. **d** The graph shows the percentage of TCR microclusters colocalized with hPD-1 2 min after contact in T cells in **c** (n = 10). **e** AND-TCR T cell hybridomas, 2D12 cells, expressing hPD-1–EGFP (green) were prestained with DyLight 650-labeled H57 Fab (red), conjugated with an MCC$_{88-103}$ prepulsed (1 μM) I-E$^k$-expressing APC line, DC-1 cell, not expressing (top) or expressing (bottom) hPD-L1-HaloTag (cyan), and real-time imaged by confocal microscopy 2 min after T cell–APC contact. All data are representative of two independent experiments. Bars, 5 μm. Data are presented as mean values ± SD. Statistical analysis was performed by an unpaired two-sided *t*-test. ****p < 0.0001. Source data for **c** and **d** are provided as a Source Data file.

conjugation assays, which showed the accumulation of SHP2 at the immune synapses correlating with the hPD-1-hPD-L1 binding (Supplementary Fig. 2a). The physical association of SHP2 with hPD-1 was also detected when AND-TCR T cells expressing hPD-1-EGFP were stimulated by MCC$_{88-103}$ peptide-prepulsed DC-1 cells expressing hPD-L1 (Fig. 2b).

We examined the phosphorylation state of the activation signaling molecules downstream of the TCR, which recruit to the hPD-1-SHP2 inhibitory microclusters. Although the TCR microclusters were highly stained by anti-phospho (p) CD3 or anti-pSH2 domain containing leukocyte protein of 76 kDa (SLP-76) at the initiation of the T cell-SLB attachment, these phosphorylated proteins were reduced if hPD-1 binds to hPD-L1 (Fig. 2c and Supplementary Fig. 2b). The ratio of pCD3 to TCR (pCD3ζ/TCR) or that of pSLP-76 to TCR (pSLP-76/TCR) was statistically decreased (Fig. 2d and Supplementary Fig. 2c). As expected, Western blotting analyses confirmed the reduction of the phosphorylation state of the further downstream molecules of a TCR/CD3 complex, phospholipase Cγ (PLCγ), and kinases Akt and Erk (Fig. 2e, f). To confirm whether these biochemical results are correlated with T cell responses, we analyzed the response of hPD-1+ AND-TCR T cells stimulated by MCC-pulsed hPD-L1+ DC-1 cells via IL-2 production measured by ELISA and found that T cells produced less IL-2 in the presence of hPD-1-hPD-L1 binding, suggesting the hPD-L1-mediated suppression of the biological responses of T cells (Fig. 2g).

## A Blocking antibody for hPD-1-hPD-L1 binding inhibits hPD-1 microcluster formation and recovers T cell suppression at a sufficient concentration of each antibody

Having confirmed the correlation between the microcluster formation of hPD-1 and the PD-1-mediated T cell suppression, we next investigated whether our imaging system with SLB would be suitable to evaluate the inhibitory effects of the anti-hPD-1 or anti-hPD-L1 used in the laboratory to that used in clinical practice, such as pembrolizumab, nivolumab, durvalumab, and atezolizumab. The avidities and optimal dose of all antibodies were examined by flow cytometry analysis with the staining of the hPD-1- or hPD-L1-expressing cells with an adequate concentration of each antibody. All antibodies were sufficient to bind to almost of all their targets at least 2 μg/ml as concentration (Supplementary Fig. 3a, b). Half maximum effective concentrations (EC$_{50}$) were calculated from the mean fluorescence intensity (MFI) of these flow cytometry data by using 4-parameter logistic function[30] and there were no significant differences in EC$_{50}$ for individual antibodies (Supplementary Fig. 3c and Supplementary Table 1). The hPD-1 microclusters formed at the interface between a T cell and an SLB expressing hPD-L1 were disrupted by the addition of anti-hPD-1/hPD-L1 antibody at sufficient concentrations (10 μg/ml) (Fig. 3a, b, Supplementary Fig. 4a, b). In the cell-cell conjugation assay, as shown in Fig. 1e, the accumulation of hPD-1 and hPD-L1 at the T cell–APC interface was disturbed by the addition of these antibodies (Fig. 3c). The amounts of IL-2 secreted from these T cells in Fig. 3c or Supplementary Fig. 4a were restored

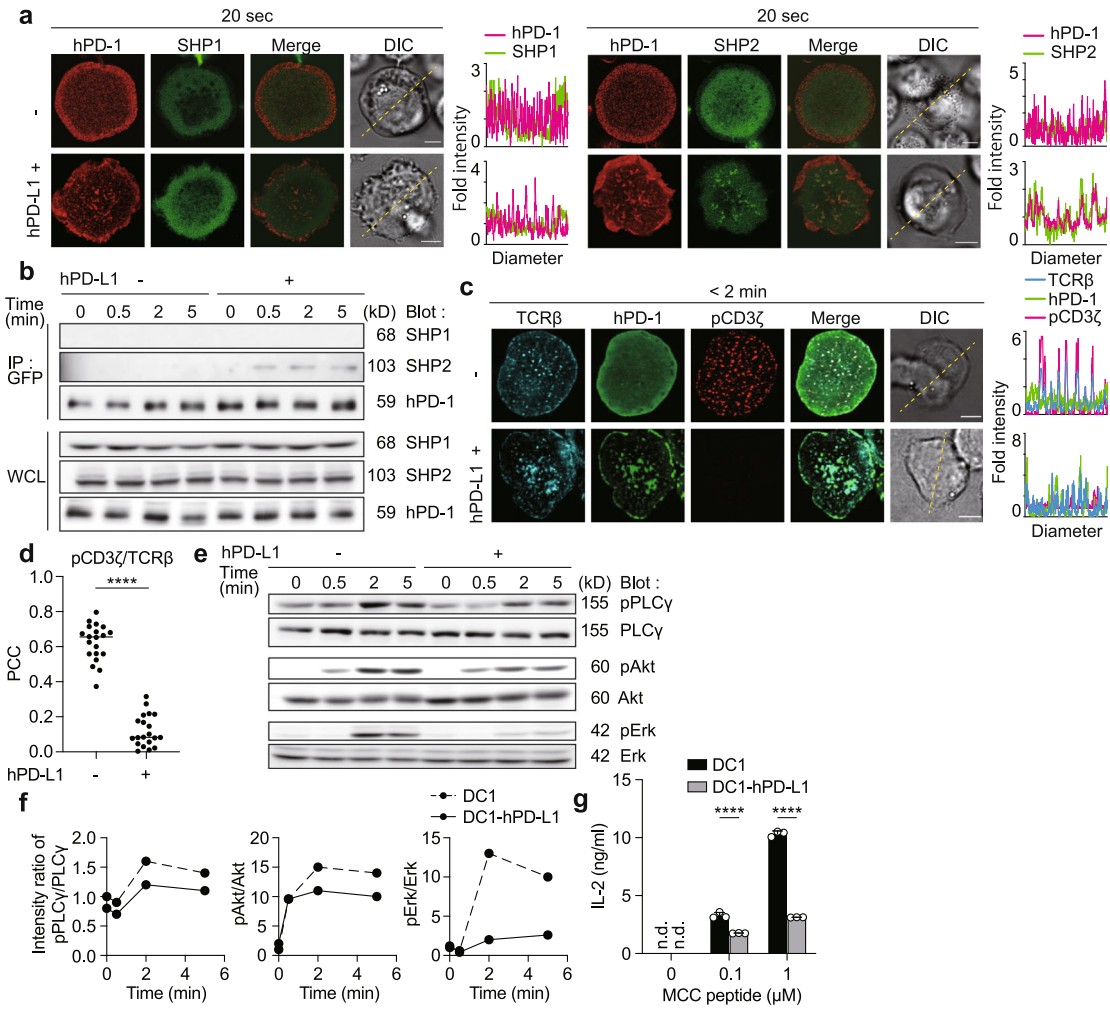

**Fig. 2 | hPD-1–hPD-L1 binding induces the recruitment of SHP2 to the hPD-1 microclusters to suppress T cell signaling by dephosphorylation of receptors and their downstream signaling molecules. a** 2D12 cells expressing both hPD-1–HaloTag (red) and EGFP–SHP1 (left, green) or –SHP2 (right, green) were plated onto the SLB without (top) or with hPD-L1–GPI (bottom). Histograms show fold fluorescent intensities of hPD-1 (magenta) and SHP1 (green) or SHP2 (green) on the diagonal yellow lines in the DIC images. **b** 2D12 cells expressing hPD-1–EGFP were conjugated by MCC$_{88-103}$-prepulsed DC-1 cells not expressing (left) or expressing hPD-L1 (right). Cells were lysed, immunoprecipitated for hPD-1 by anti-GFP, and blotted for SHP1, SHP2, or hPD-1. The WCLs were blotted for SHP1, SHP2, or GFP. **c** 2D12 cells expressing hPD-1–EGFP (green) were prestained with DyLight 549-labeled H57 Fab (cyan), plated onto an SLB without (top) or with hPD-L1–GPI (bottom), fixed at 2 min after contact, stained with anti-phospho (p) CD3ζ (red). Histograms show fold fluorescent intensities of TCRβ (cyan), hPD-1(green),

and pCD3ζ (magenta) on the diagonal yellow lines in the DIC images. **d** The scatter plot summarizing the Pearson's colocalization coefficients (PCC) values in (**c**). PCC was calculated between pCD3ζ/TCRβ in the absence (left, 0.6232 ± 0.05) or presence (right, 0.1198 ± 0.05) of hPD-L1–GPI by 20 randomly plotted profiles on 20 cells. **e** The cells in **b** were stimulated with MCC$_{88-103}$-prepulsed DC-1 cells not expressing (left) or expressing hPD-L1 (right). The WCLs were blotted for pPLCγ1, PLCγ1, pAkt, Akt, pErk1/2 or Erk1/2. **f** The graphs show the intensity ratio of pPLCγ/PLCγ (left), pAkt/Akt (middle) or pErk/Erk (right) in (**e**). **g** The cells in **b** were cocultured with DC-1 cells (black) or those expressing hPD-L1 (gray) plus indicated concentrations of MCC$_{88-103}$. The concentration of IL-2 was measured by ELISA. n.d. not detected. All data are representative from two independent experiments. Bars, 5 μm. Data are presented as mean values ± SD. Statistical analysis was performed by unpaired two-sided *t*-test and one-way ANOVA. ****$p$ < 0.0001. Source data for **a**–**g** are provided as a Source Data file.

by the addition of anti-hPD-1/hPD-L1 antibodies (Fig. 3d and Supplementary Fig. 4c). We prepared splenic CD8$^+$ T cells from OT-I TCR Tg (specific for Ovalbumin 257-264 [OVA$_{257-264}$] loaded on H-2K$^b$) Rag2-deficient (*Rag2$^{-/-}$*) PD-1-deficient (*Pdcd1$^{-/-}$*) mice, reconstituted by hPD-1 (Supplementary Fig. 4d), and examined the cytotoxicity of these hPD-1-expressing T cells by coculture with the H-2K$^b$ tumor cell line, EL-4 cells, without or with hPD-L1 expression (Supplementary Fig. 4e). OT-I Tg *Rag2$^{-/-}$ Pdcd1$^{-/-}$* CD8$^+$ T cells expressing hPD-1 showed less cytotoxicity if these cells were cocultured by EL-4 cells expressing hPD-L1, but the suppression mediated by hPD-1-hPD-L1 binding was restored in the addition of anti-hPD-1 or anti-hPD-L1 (Fig. 3e). We further evaluated the blocking effects of these antibodies on PD-1-PD-L1 interaction by our imaging system using hPD-1-EGFP-expressing OT-I Tg T cells and SLBs reconstituted by hPD-L1. Microcluster

formation of hPD-1 was inhibited by the addition of anti-hPD-1 or anti-hPD-L1 antibodies as in cytotoxic assays (Supplementary Fig. 4f, g).

We also evaluated the production of IFNγ by hPD-1-GFP$^+$ OT-I Tg T cells by stimulating with the HCC827 cells, expressing H-2K$^b$ and upregulating hPD-L1 as shown in Supplementary Fig. 1c. The amounts of IFNγ from these T cells tended to be restored by the addition of anti-hPD-1/hPD-L1 antibodies (Supplementary Fig. 4h). In the cell-cell conjugation experiments using these cells, hPD-1 was accumulated at the T cell-HCC827 cell interface, and that was disrupted by the addition of antibodies (Supplementary Fig. 4i, j).

These results clearly showed a positive correlation between the inhibition of PD-1 microcluster formation and the reverse effect of T cell response inhibition, such as the reduction of IL-2 production and cytotoxic function, by adding anti-hPD-1 or anti-hPD-L1.

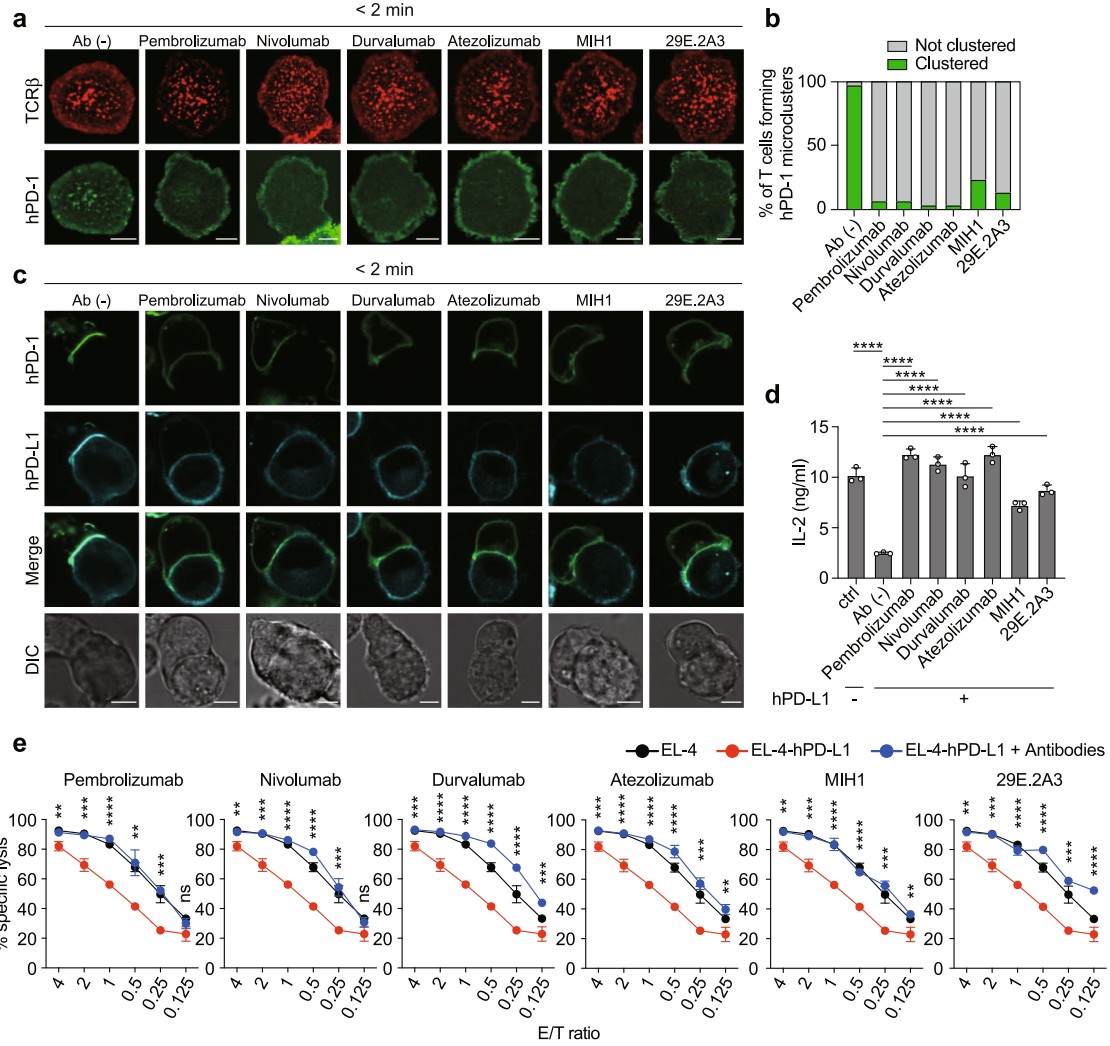

**Fig. 3 | The inhibition of hPD-1 microcluster formation is correlated with the recovery of T cell responses if blocking antibodies for hPD-1 or hPD-L1 were added in sufficient concentrations. a** 2D12 cells expressing hPD-1-EGFP (green) were prestained with DyLight 650-labeled H57 Fab (red) and plated on an SLB with hPD-L1–GPI as shown in Fig. 1a. The cells were imaged by confocal microscopy 2 min after contact in the absence (top) or presence of pembrolizumab (anti-hPD-1, row 2), nivolumab (anti-hPD-1, row 3), durvalumab (anti-hPD-L1, row 4), atezolizumab (anti-hPD-L1, row 5), MIH1 (anti-hPD-L1, row 6), or 29E.2A3 (anti-hPD-L1, bottom) at a concentration of 10 μg/ml. **b** The graph shows the percentage of T cells forming hPD-1 microclusters in **a** ($n = 30$). **c** The cells in **a** were conjugated with MCC88-103 prepulsed (1 μM) DC-1 cells expressing hPD-L1-HaloTag (cyan) in the absence or presence of the indicated antibodies and images by confocal microscopy 2 min after T cell–APC contacts. **d** The cells in **a** were cocultured for 16 h with

1 μM MCC88-103 and DC-1 cells not expressing or expressing hPD-L1 in the absence or presence of the indicated antibodies, and the concentration of IL-2 in each supernatant was measured by enzyme-linked immunosorbent assay (ELISA) in triplicate. **e** Pdcd1⁻/⁻ OT-I TCR-Tg T cells reconstituted by hPD-1 were cocultured with 1 nM OVA257-264-pulsed target EL-4 cells not expressing or expressing hPD-L1 at the indicated E/T ratios for 16 h without or with anti-hPD-1 or anti-hPD-L1 at a concentration of 10 μg/ml. Statistics were performed between the calculated percentage of specific lysis with and without each antibody. All data are representative of two independent experiments. ns not significant. Bars, 5 μm. Error bars, SD. Statistical analysis was performed by one-way ANOVA. *$p < 0.05$, **$p < 0.01$, ***$p < 0.001$, ****$p < 0.0001$. Source data for **b**, **d** and **e** are provided as a Source Data file.

## Each antibody requires its own optimal concentration for recovering from T cell suppression

We next examined the minimal concentrations of various blocking antibodies against hPD-1-hPD-L1 binding as described above. The formation of hPD-1 microclusters was uniformly blocked by all four antibody clones at a concentration of 10 μg/ml, but each clone possesses its own minimal concentration required for the inhibition of hPD-1 microcluster formation (Fig. 4a). These imaging data were analyzed to calculate the ratio of the total area of microclusters inside T cells (%area) by dividing the summation of the areas of hPD-1 microclusters by the total cell area and to confirm those concentrations required for each antibody (Fig. 4b, Supplementary Fig. 5a, b). To compare the minimal dose of each antibody to block hPD-1 clustering with that to suppress the T cell biological response, we measured the

concentration of IL-2 produced by hPD-1-expressing T cells further treated with the same blocking antibody shown in Fig. 3d (Fig. 4c). We then performed a linear regression analysis and confirmed a strong negative correlation between the total area of hPD-1 microclusters in Fig. 4b and the IL-2 production in Fig. 4c, indicating that imaging analysis could be suitable for evaluating biological functions (Fig. 4d). By calculating the percent effect concentrations (Table 1) from the dose-response curves (Fig. 4e and Supplementary Fig. 5c)[30], we verified whether the calculated percent effect concentrations were correlated with the imaging data described above and drew out the rates for blocking hPD-1 microcluster formation at EC25, EC50, and EC75 for each clone. We noticed that the EC50 rates of all four clones were sufficient to block hPD-1 microcluster formation in a majority of the T cells (Fig. 4f). Similar results were obtained from the biological examination

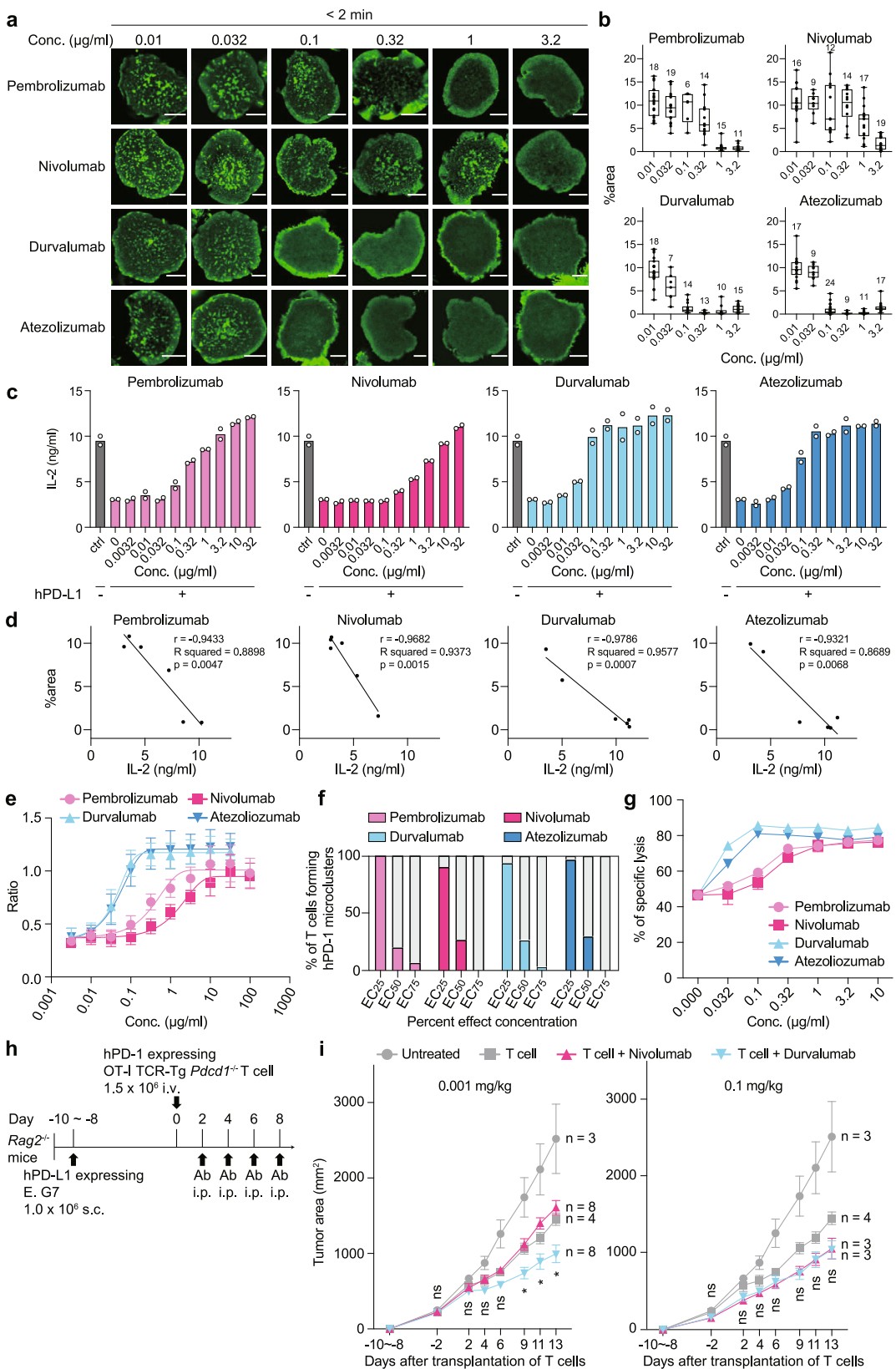

of what concentrations of antibodies were required to restore the cytotoxicity by *Pdcd1⁻/⁻* OT-I TCR-Tg CD8⁺ primary T cells expressing hPD-1 (Fig. 4g, Supplementary Fig. 5d, e and Supplementary Table 2). To evaluate these in vitro findings by murine in vivo experiments, we applied a tumor-bearing mice model in which we further introduced hPD-L1 into the OVA-transfected lymphoma cell line EL-4, E.G7

(Supplementary Fig. 5f), and transplanted E.G7-hPD-L1 into *Rag2⁻/⁻* albino C57BL/6 mice with *Pdcd1⁻/⁻* OT-I TCR-Tg CD8⁺ primary T cells reconstituted by hPD-1. These tumor-bearing mice were inoculated with every ICI at an indicated concentration and the tumor size in each mouse was then measured to compare between various groups so as to judge the effects on tumor reduction (Fig. 4h, i and Supplementary

**Fig. 4 | ICIs require their own concentration optimal for blocking hPD-1–hPD-L1 binding. a** 2D12 cells expressing hPD-1-EGFP (green) were plated on an SLB with hPD-L1–GPI. The cells were imaged 2 min after contact with each antibody at the indicated concentrations. Bars, 5 μm. **b** The ratio of the total area of microclusters to the area of the T cell-SLB interface in individual T cells (%area) in (**a**). Data were analyzed from the total number of cells indicated above the boxes. Box center indicates median, box edges 25th and 75th percentile, and whiskers minimum and maximum percentile. **c** The cells in **a** were cocultured with MCC$_{88-103}$ and DC-1 cells not expressing or expressing hPD-L1 in the absence or presence of each antibody at the indicated concentrations, and the concentration of IL-2 was measured by enzyme-linked immunosorbent assay (ELISA). **d** PCC, R squared, *p* value and linear regression equation between the results of (**b**) and (**c**). **e** The data in **c** were normalized among four anti-hPD-1 and anti-hPD-L1. T cell response curve was depicted by a 4-parameter logistic function. Results represent summarized data from six

independent experiments. Each plot is the average of these experiments. **f** The percentages of T cells forming hPD-1 microclusters in the presence of each antibody at the indicated percent effective concentrations (*n* = 30). **g** The cytotoxicity of the effector cells, *Pdcd1*$^{-/-}$ OT-I TCR-Tg T cells reconstituted by hPD-1, against the target cells, OVA$_{257-264}$-prepulsed EL-4 cells, at the E/T ratio of 1 in the presence of each antibody. **h** The time course of tumor-bearing mice experiments in vivo. **i** The graphs show the growth curves of hPD-L1-expressing E.G7 Cells in *Rag2*$^{-/-}$ mice further injected with *Pdcd1*$^{-/-}$ OT-I TCR-Tg CD8$^+$ T cells plus nivolumab or durvalumab at a concentration of 0.001 mg/kg (left) or 0.1 mg/kg (right), as shown in **h**. ns, not significant. Bars, mean ± SEM. All data are representative of two independent experiments. Error bars, SD, except for (**i**). Statistical analysis was performed by one-way ANOVA. *$p < 0.05$. Source data for **b**–**g** and **i** are provided as a Source Data file.

**Table 1 | The concentration of the percent effects, EC$_{20}$, EC$_{25}$, EC$_{50}$, EC$_{75}$, EC$_{90}$ and EC$_{98}$ of pembrolizumab, nivolumab, durvalumab or atezolizumab required for the recovery from the PD-1-mediated suppression of IL-2 production from T cells**

| | Pembrolizumab | Nivolumab | Durvalumab | Atezolizumab (μg/ml) |
|---|---|---|---|---|
| EC$_{20}$ (95% CI) | 0.07804 | 0.3389 | 0.02113 | 0.02095 |
| | 0.02118–0.1680 | 0.1390–0.6495 | 0.01163–0.03035 | 0.008643–0.03598 |
| EC$_{25}$ (95% CI) | 0.1074 | 0.4734 | 0.02431 | 0.02511 |
| | 0.03481–0.2143 | 0.2217–0.8497 | 0.01444–0.03383 | 0.01155–0.04090 |
| EC$_{50}$ (95% CI) | 0.3629 | 1.695 | 0.04146 | 0.05018 |
| | 0.1981–0.6492 | 0.9932–3.329 | 0.03134–0.05420 | 0.03306–0.07170 |
| EC$_{75}$ (95% CI) | 1.227 | 6.07 | 0.07073 | 0.1003 |
| | 0.6103–3.189 | 2.893–21.44 | 0.04732–0.1028 | 0.06622–0.1632 |
| EC$_{90}$ (95% CI) | 4.146 | 21.74 | 0.1207 | 0.2004 |
| | 1.423–20.92 | 7.264–154.3 | 0.06179–0.2261 | 0.1061–0.4603 |
| EC$_{98}$ (95% CI) | 27.13 | 155.5 | 0.275 | 0.5828 |
| | 4.828–434.9 | 28.01–3412 | 0.09277–0.8269 | 0.2097–2.582 |

The data were calculated by the dose–response curves from the experiments in Fig. 4e.

Fig. 5g). In the groups receiving nivolumab or durvalumab at high concentrations (0.1 mg/kg), both ICIs showed similar suppressive effects on the growth of the transplanted E.G7-hPD-L1 (Fig. 4i, right). However, durvalumab, but not nivolumab, possessed sufficient capacity to reduce the tumor growth in each group receiving an ICI at a low concentration (0.001 mg/kg) (Fig. 4i, left).

Additionally, we evaluated the inhibitory effects of each antibody on hPD-1 microcluster formation at different hPD-L1 densities on SLBs. It was found that even the densities were lower than 150 molecules/μm$^2$, such as 75 molecules/μm$^2$ and 37.5 molecules/μm$^2$, it was possible to confirm the differences in blocking ability of each antibody (Supplementary Fig. 6a, b).

These results indicate that each ICI has an optimal concentration to restore T cell function through blocking hPD-1-hPD-L1 binding both in vitro and in vivo, and it is useful to determine the optimal concentrations of each ICI to image the hPD-1 microclusters as an indicator for T cell exhaustion via hPD-1.

## hPD-L2 possesses the same inhibitory functions as hPD-L1 and is useful to determine the actual effects of ICIs

Since our imaging system could be adapted to evaluate the effects of ICIs on some cancer types expressing hPD-L1, we next established an experimental system using an SLB reconstituted by hPD-L2 to examine tumors expressing hPD-L2. hPD-L2 could also introduce the accumulation of hPD-1 at TCR microclusters on an SLB loaded by hPD-L2-GPI with the same kinetics as in hPD-L1 (Fig. 5a, b). The formation of the hPD-1 microcluster triggered by hPD-1-hPD-L2 binding was inhibited by the addition of anti-hPD-1 or anti-hPD-L2 antibody, 24 F.10C12, at a concentration of 10 μg/ml (Fig. 5c, d and Supplementary Fig. 7a). In the

cell-cell conjugation experiments, both hPD-1 and hPD-L2 were accumulated at the T cell-DC-1 cell interface only if DC-1 cells expressed hPD-L2 (Supplementary Fig. 7b, c), and the accumulation of hPD-1 and hPD-L2 was disrupted by the addition of anti-hPD-1 or anti-hPD-L2 (Fig. 5e). We measured the production of IL-2 by hPD-1-expressing T cells by stimulation with the hPD-L2$^+$ APCs shown in Fig. 5e to evaluate the biological response under the control of hPD-1-hPD-L2 binding. The hPD-L2$^+$ DC-1 cells distinctly reduced the IL-2 production by hPD-1$^+$ T cells, and this hPD-L2-mediated suppression was canceled by the addition of anti-hPD-1 or anti-hPD-L2 (Fig. 5f). Based on the dose-response curves depicted from the ecoefficiency between the concentration of each antibody and the IL-2 production by these T cells (Fig. 5g, Supplementary Fig. 7d, e), the percent effect concentrations were calculated (Supplementary Table 3). As we noticed in the case of hPD-L1, the concentration of blocking antibodies required to inhibit the hPD-1-hPD-L2 production was different for each antibody.

To mimic a clinical situation of a tumor expressing both PD-L1 and PD-L2 for which a clinician could not decide which ICI was suitable, we reconstituted both hPD-L1- and hPD-L2-GPI proteins onto an SLB and imaged the behavior of hPD-1 on AND-TCR-expressing T cells. On an SLB expressing hPD-L1 and hPD-L2, the T cells formed rigid hPD-1 microclusters that were disrupted by the addition of anti-hPD-1 or anti-hPD-L1 plus anti-hPD-L2 (Fig. 6a, b). These hPD-1 microclusters were stabilized even if only anti-hPD-L1 or anti-hPD-L2 was added to T cells expressing hPD-1. The administration of anti-hPD-1 alone or the combination of anti-hPD-L1 and anti-hPD-L2, but not the solo use of anti-hPD-L1 or anti-hPD-L2, interfered with the accumulation of hPD-1, hPD-L1 and hPD-L2 at the interface between an hPD-1$^+$ T cell and a DC-1 cell expressing both hPD-L1 and hPD-L2 in the cell-cell conjugation assay

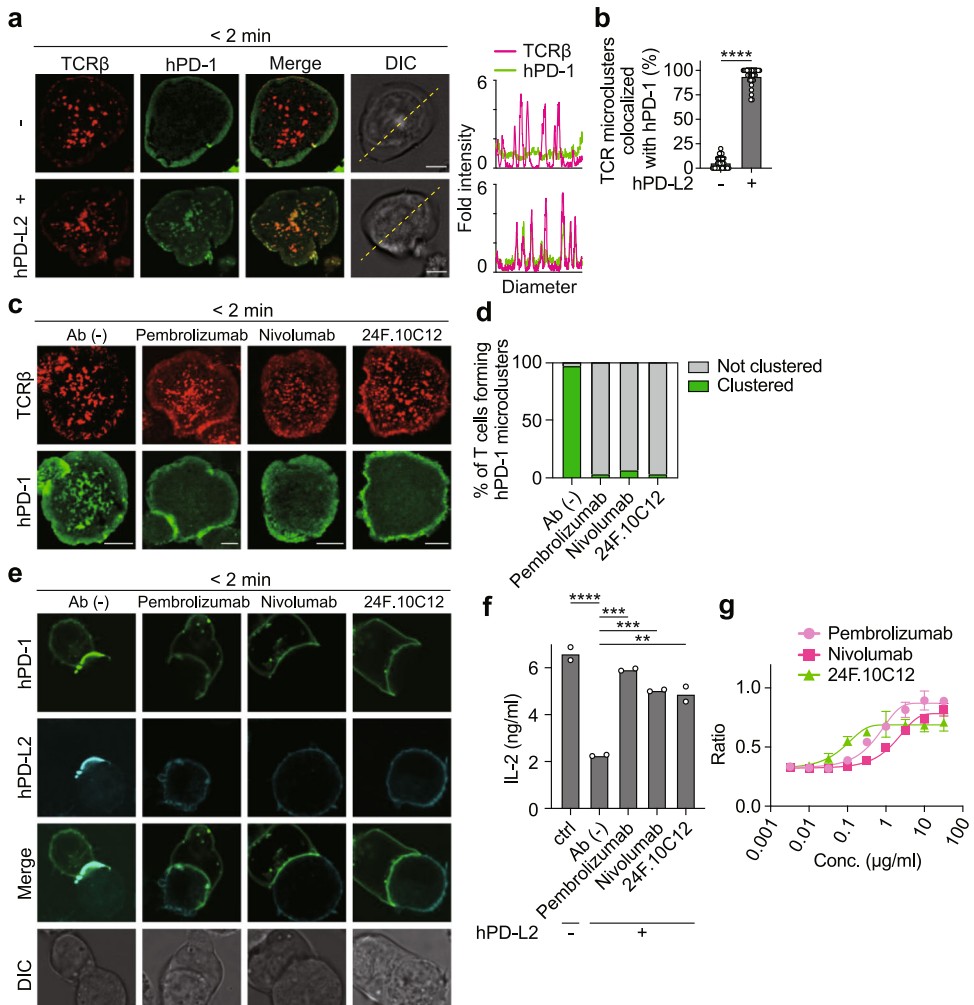

**Fig. 5 | hPD-L2 forms hPD-1 microclusters to suppress T cell response in a similar fashion as hPD-L1. a** 2D12 cells expressing hPD-1-EGFP (green) were imaged as shown in Fig. 1c on an MCC$_{88-103}$-prepulsed SLB containing I-E$^k$− and mICAM-1−GPI without (top) or with hPD-L2−GPI (bottom). TCRβ, red. Histograms show fold fluorescent intensities of TCRβ (magenta) and hPD-1 (green) on the diagonal yellow lines in the DIC images. **b** The graph shows the percentages of TCR microclusters colocalized with hPD-1 2 min after contact in **a** (*n* = 30). **c** The cells in **a** were imaged in the absence (top) or presence of pembrolizumab (anti-hPD-1, row 2), nivolumab (anti-hPD-1, row 3), or 24 F.10C12 (anti-hPD-L2, bottom) at a concentration of 10 μg/ml. **d** The graph shows the percentages of T cells forming hPD-1 microclusters in **c** (*n* = 30). **e** The cells in **a** were conjugated with MCC$_{88-103}$ prepulsed (1 μM) DC-1 cells expressing hPD-L2-HaloTag (cyan) in the absence or presence of the indicated

antibodies at a concentration of 10 μg/ml and imaged 2 min after contacts. **f** The cells in **a** were cocultured for 16 h with 1 μM MCC$_{88-103}$ and DC-1 cells not expressing or expressing hPD-L2 in the absence or presence of each antibody at a concentration of 10 μg/ml. Concentration of IL-2 in each supernatant was measured by enzyme-linked immunosorbent assay (ELISA). **g** Data in **f** were normalized among three antibodies and T cell response curves were depicted by a 4-parameter logistic function, as shown in Fig. 4e. Results are the summarized data from three independent experiments. Each plot is the average of these experiments. All data are representative of two independent experiments. Bars, 5 μm. Data are presented as mean values ± SD. Statistical analysis was performed by an unpaired Student's *t*-test and one-way ANOVA. **p < 0.01, ***p < 0.001, ****p < 0.0001. Source data for **a**, **b**, **d**, **f** and **g** are provided as a Source Data file.

(Fig. 6c) and simultaneously restored the IL-2 production from such T cells in the same experimental set (Fig. 6d).

### A tendency might be found to cancel the PD-1-mediated T cell suppression more effectively with combinational use of anti-hPD-1 and anti-hPD-1 ligands

To determine the more effective combinations among four ICIs and their optimal doses in such usages, we examined the responses of the AND-TCR T cells expressing hPD-1 stimulated by MCC$_{88-103}$-prepulsed DC-1 cells expressing hPD-L1 in the presence of various combinations and doses of ICIs. As shown in Table 1, compared to EC$_{90}$ and EC$_{98}$, the EC$_{50}$ required to restore IL-2 production from these T cells was significantly low for every antibody. Nevertheless, the combinational use of anti-hPD-1 and anti-hPD-L1 at each EC$_{50}$ showed greater enhancement of IL-2 production than a single use of each antibody at its own

EC$_{90}$ (Fig. 7a, b). A combination of the different clones of anti-hPD-1 or anti-hPD-L1 was not as effective as those of anti-hPD-1 and anti-hPD-L1 in the restoration of IL-2 production. Furthermore, such a combined effect of simultaneous use of anti-PD-1 and anti-PD-L1 in T cell inhibition were evaluated by imaging whether the formation of hPD-1 microcluster was inhibited. When anti-hPD-1 or anti-hPD-L1 was solely added at a low concentration of EC$_{20}$, or when two different clones of anti-hPD-1 or anti-hPD-L1 are added, hPD-1 microclusters were remained, but if T cells are treated by a combination of anti-hPD-1 and anti-hPD-L1 at a low concentration of each EC$_{20}$, the formation of hPD-1 microclusters was definitely inhibited (Fig. 7c, d). As shown in Fig. 4b, the total areas of hPD-1 microclusters were reduced when anti-hPD-1 and anti-hPD-L1 were added in combination (Fig. 7e).

Finally, the effect of the combination treatment with three antibodies, anti-hPD-1, hPD-L1, and hPD-L2, was evaluated by the same

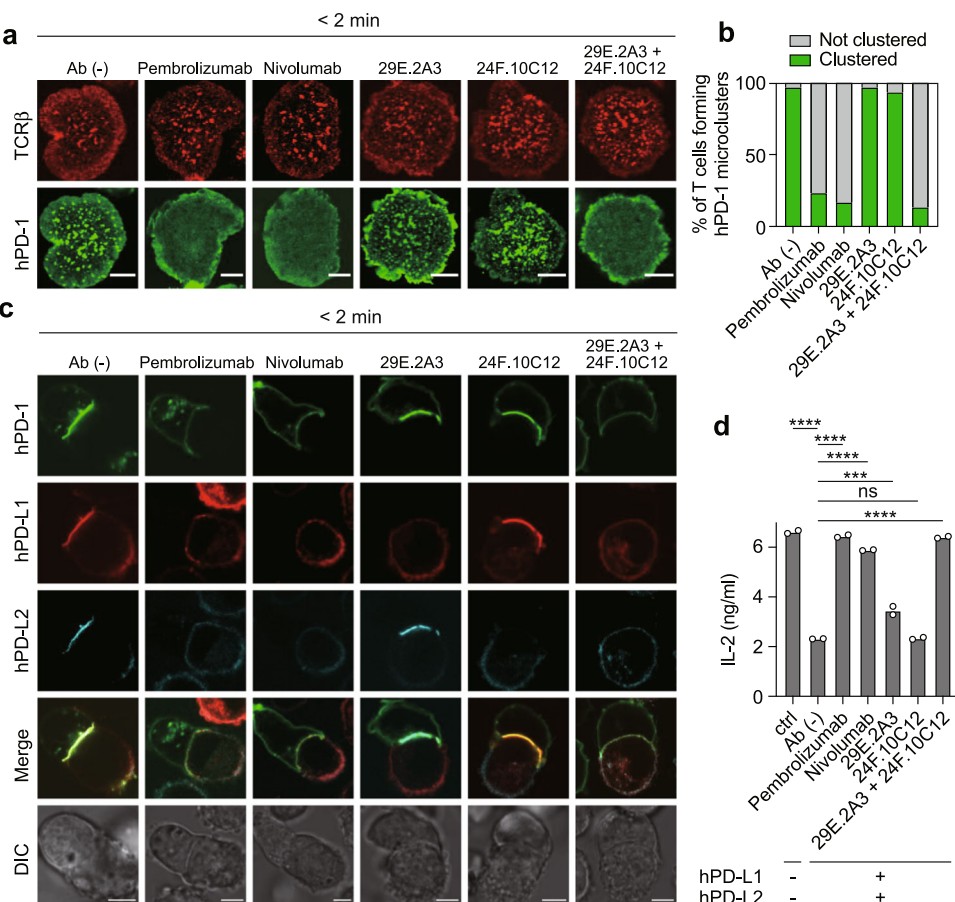

**Fig. 6 | Anti-hPD-1 solely blocks, but anti-hPD-L1 and anti-hPD-L2 cooperatively block the binding between hPD-1 and hPD-1 ligands. a** The cells were imaged as shown in Fig. 1c 2 min after contact to an MCC$_{88-103}$-prepulsed SLB containing I-E$^{k}$–, mICAM-1–, hPD-L1–, and hPD-L2–GPI in the absence (top) or presence of pembrolizumab (anti-hPD-1, row 2), nivolumab (anti-hPD-1, row 3), 29E.2A3 (anti-hPD-L1, row 4), 24 F.10C12 (anti-hPD-L2, row 5), or both 29E.2A3 and 24 F.10C12 (bottom) at a concentration of 10 µg/ml. hPD-1-EGFP, green; TCR, red. **b** The graph shows the percentages of T cells forming hPD-1 microclusters in **a** ($n = 30$). **c** The cells in **a** were conjugated with an MCC$_{88-103}$-prepulsed (1 µM) DC-1 cells expressing both hPD-L1-SNAP-tag (red) and hPD-L2-HaloTag (cyan) plus antibodies as in **a** and images 2 min after contacts. **d** The cells in **a** were cocultured for 16 h with 1 µM MCC$_{88-103}$ and DC-1 cells not expressing or expressing both hPD-L1 and hPD-L2 plus antibodies as in (**a**). IL-2 in each supernatant was measured by enzyme-linked immunosorbent assay (ELISA). ns not significant. All data are representative of three independent experiments. Bars, 5 µm. Data are presented as mean values ± SD. Statistical analysis was performed by one-way ANOVA. ***$p < 0.001$, ****$p < 0.0001$. Source data for **b** and **d** are provided as a Source Data file.

biological assay as shown in Fig. 7a, b, where each combination of antibodies was added to the aggregation cultures of hPD-1$^{+}$ AND-TCR T cells with MCC$_{88-103}$-prepulsed DC-1 cells co-expressing hPD-L1 and hPD-L2. When anti-hPD-L1 or anti-hPD-L2 was added alone, less recovery of IL-2 production from T cells was observed at all antibody concentrations, but it was under 10 µg/ml. However, the recovery of IL-2 production was observed to be dependent on the concentration of antibody when anti-hPD-1 or two or more antibodies were used in combination. If the triple antibodies, anti-hPD-1, anti-hPD-L1, and anti-hPD-L2, were used in combination, each antibody was sufficient at quite low concentration, 0.1 µg/ml, for the recovery of IL-2 production (Fig. 7f). Translocation of hPD-1, hPD-L1, and hPD-L2 at the T cell-DC-1 cell interface was completely blocked in the presence of triple antibodies, each at low concentration, while at least one of hPD-1, hPD-L1, and hPD-L2 remained at the interface if antibodies were used in a single or double combination (Fig. 7g, h). Based on these results, we suggest that the combinational use of antibodies against multiple targets would be more effective with each at low dose compared to solo antibody use at high dose.

## Discussion

In this report, we established a single-molecule imaging system to visualize the precise behaviors of the human immune checkpoint receptor, hPD-1, and rediscovered a signalosome, hPD-1 microcluster, formed by the binding of hPD-1 to its ligands, hPD-L1 and hPD-L2. We further confirmed in detail the recruitment of a phosphatase, SHP2, at PD-1 microclusters to attenuate TCR signaling by dephosphorylation of the signaling molecules in the downstream of TCRs and fully defined the hPD-1 microcluster as an inhibitory signalosome by various methods of biochemistry and physiology. We evaluated the blocking effects of each clone of anti-PD-1, anti-PD-L1, or anti-PD-L2 in clinical use and confirmed that our molecular imaging strategy was correlated with the conventional analyses of biological functions.

In clinical practice, the same amount of ICIs or weight-adjusted doses of ICIs are uniformly administered to patients with different weights and general health conditions, and therefore it is difficult to judge whether the blood concentration is optimal for each patient. We confirmed that hPD-1-hPD-L1 or -hPD-L2 binding was uniformly inhibited by the addition of anti-hPD-1, anti-hPD-L1, or anti-hPD-L2 at sufficient concentrations, but each antibody possessed its own minimal concentration required to physically inhibit the structural binding and to recover the T cells from exhaustion status physiologically. Similar results were obtained from a murine tumor-bearing mice model in vivo, in which OVA-expressing lymphoma cell line E.G7 cells were transplanted and introduced antitumor responses by CD8$^{+}$ T cells from OT-I TCR-Tg mice. The differences in EC$_{50}$ values from the previous

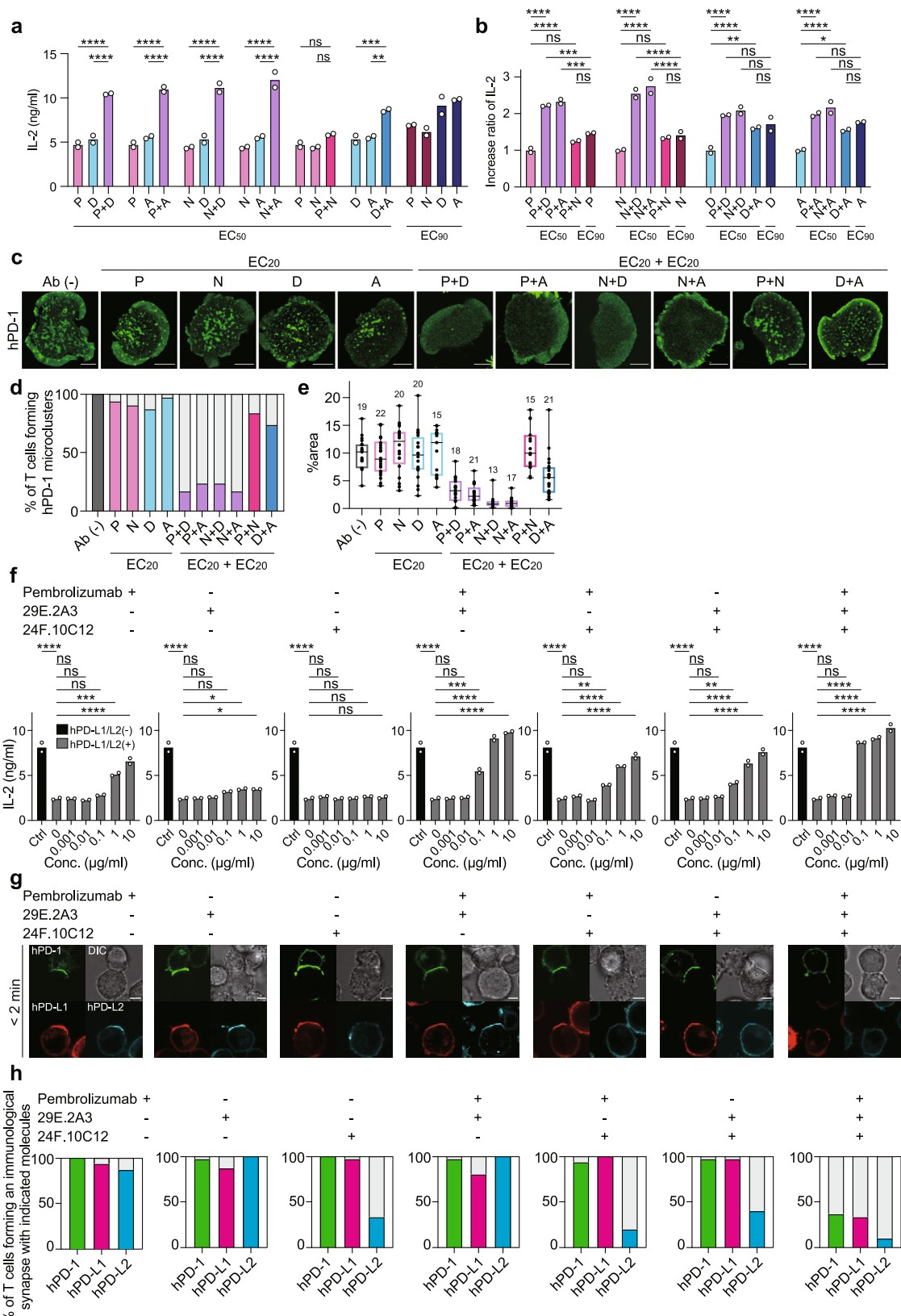

report[30] may be influenced by the expression levels of hPD-1 or hPD-1 ligands. Furthermore, in this study, cells derived from mice were used, but in order to assess the blocking ability of therapeutic antibodies, it would be better to approximate the human system, which is the next our research task.

We also found that the combined usage of anti-PD-1 and anti-PD-L1 more efficiently recovered hPD-1-mediated T cell suppression than the usage of two clonal antibodies whose targets are the same, PD-1 or PD-L1. Furthermore, we evaluated the effects of the antibody cocktail on T cell recovery using triple antibodies that contain anti-PD-1, anti-PD-L1,

**Fig. 7 | The combinational use of anti-hPD-1 and anti-hPD-1 ligands more effectively cancels hPD-1-meidated T cell suppression. a** 2D12 cells expressing hPD-1-EGFP were cocultured with $MCC_{88-103}$ and DC-1 cells expressing hPD-L1 in the presence of the indicated combination of antibodies at an individual concentration calculated from $EC_{50}$. The concentration of IL-2 was measured by enzyme-linked immunosorbent assay (ELISA). P, pembrolizumab; N, nivolumab; D, durvalumab; A, atezolizumab. **b** The ratios of the increment of IL-2 measured in (**a**). **c** The cells in **a** were imaged at 2 min after contact to an $MCC_{88-103}$-prepulsed SLB containing I-E$^k$−, mICAM-1−, and hPD-L1−GPI in the absence or presence of the indicated combination of antibodies at an individual concentration calculated from $EC_{20}$. **d** The percentages of T cells forming hPD-1 microclusters in (**c**) ($n = 30$). **e** The graph shows the ratio of the total area of microclusters to the area of the T cell-SLB interface in individual T cells (%area) in (**c**). Data were analyzed from the total number of cells indicated above the boxes. Box center indicates median, box edges 25th and 75th percentile, and whiskers minimum and maximum percentile. **f** The same coculture as in **a** was performed using DC-1 cells expressing both hPD-L1 and hPD-L2 in the absence or presence of the antibodies at the indicated concentrations. **g** The cells in **a** were conjugated with $MCC_{88-103}$-prepulsed DC-1 cells expressing both hPD-L1-SNAP-tag (red) and hPD-L2-HaloTag (cyan) in the absence or presence of pembrolizumab and/or 29E.2A3 and/or 24 F.10C12 at a concentration of 0.1 μg/ml. The images were acquired 2 min after contacts. **h** The graphs show the percentages of the cells accompanied by the accumulation of hPD-1 (green), hPD-L1 (magenta), or hPD-L2 (cyan) at the T cell-DC-1 cell interface ($n = 30$). All data are representative of two independent experiments. ns not significant. Bars, 5 μm. Error bars, standard deviation (SD). Statistical analysis was performed by one-way ANOVA. *$p < 0.05$, **$p < 0.01$, ***$p < 0.001$, ****$p < 0.0001$. Source data for **a**, **b**, **d**–**f** and **h** are provided as a Source Data file.

and anti-PD-L2, and we found a small but reliable advantage in the usage of multiple clones against different targets. As an example of the effective usage of multiple antibodies, Qiu et al. reported that a cocktail consisting of three different monoclonal antibodies targeting the Ebola glycoprotein, optimized from previous antibody cocktails, is a successful therapy to treat infections caused by Ebola virus[31]. In our experiments, we found little efficacy for the combinational use of antibodies against the same target, but we may obtain greater efficacy if we use other antibodies whose epitopes are different. We believe that selecting the best antibody combinations is important to effectively treat certain diseases.

Previous reports have shown that PD-1 strongly recruits SHP2, but not SHP1, in both T cells and B cells[14,27,32]. On the other hand, some reports discussed that PD-1 recruits both SHP1 and SHP2 in T cells[33]. Although we could not confirm the recruitment of mSHP1 to hPD-1 in our experiments, we could not deny a possibility that mSHP1 may also act in a similar way as mSHP2. Furthermore, we have shown that the recruitment of SHP2 to PD-1 microclusters occurs transiently and quickly after mPD-1 is crosslinked by mPD-L1 or mPD-L2. The clustering of SHP2 would take just tens of seconds[14,27]. In contrast, other reports have biochemically demonstrated that human SHP2 was immediately recruited to the cytoplasmic tail of hPD-1, while the colocalization of hSHP2 at hPD-1 tends to be more prolonged than mPD-1 with mSHP2[34]. In this report, we also confirmed a prolonged association of SHP2 with hPD-1 maintained from 30 s to 5 min after antigen stimulation. The difference in the duration of the PD-1-SHP2 association between murine and human PD-1 could be explained by structural differences, particularly in the amino acid sequences in or around the ITIM and/or the ITISM[32,33], so further examinations are required to prove this consideration.

In recent years, numerous ICIs have been under development all over the world. Each company is emphasizing the distinctive characteristics of their products from others, such as the isoforms of the antibodies, including the presence or absence of antibody-dependent cell-mediated cytotoxicity, the chimerism of the humanized antibodies between human and other animals, and the structural stabilities based on the features analyzed by protein science, crystallography, molecular engineering, and so on. However, as mentioned above, which patient is indicated for each ICI and which is the best for that patient is at present simply based on the first design of the phase I trial. Since each indication of ICI for each patient is determined by their age and medical history of any prior therapies, there is currently no scientific basis for the indications. We first confirmed that the suppressive status of T cells can be equally cancelled by adding anti-hPD-1, anti-hPD-L1, or anti-hPD-L2 at a sufficient concentration of antibodies. We next found that these antibodies require their own optimal concentrations for recovering from T cell suppressiveness. Clearly, every ICI should be administered to a patient at an appropriate dosage, with consideration given to the difference between the patient's condition and the experimental settings in the clinical trials. If the efficacy is similar, it

should usually be recommended to administer the minimal dose of antibodies. As such, we believe that the dose of the ICIs should be reduced to the appropriate amount as a reasonable goal. Of course, the areas of antibody-binding sites are not usually correlated with the affinities of those antibodies, but if these areas are significantly more overlapped with the sites to which those antibodies bind, such antibodies will provide more advantages in the blocking of receptor-ligand pairs[35–38]. Here, we suggest that the differences in the dosages of the antibodies required for recovering from PD-1-mediated T cell suppression are possibly due to the differences in the epitopes to which those antibodies bind. Since hPD-L1 are expressed on tumor cells at various levels in different histological classifications or tumor microenvironments, the inability to precisely set the number of hPD-L1 molecules on an SLB the same as those expressed on tumors may cause a discrepancy in which the majority of hPD-1 microclusters are inhibited to form by the addition of every ICI at concentrations of $EC_{50}$. We hope that we can evaluate the effects of ICIs on tumor suppression more accurately if we mimic the experimental conditions with SLBs more closely to the setting of dendritic cells in the draining lymph nodes of tumor-bearing patients or in the tumor microenvironment.

It is important to note that the formation of hPD-1 microclusters cannot be observed if the density of hPD-L1 on SLB is deviated from a certain range. As shown in Supplementary Fig. 1a and Supplementary Fig. 7a, b, hPD-1 microcluster formation and its inhibition by adding antibodies can be observed when the density of hPD-L1 is between 37.5 molecules/μm² and 150 molecules/μm², but it is difficult to visualize them when the density is below 18.75 molecules/μm² or above 300 molecules/μm². The development of an imaging system with a wider range of hPD-L1 molecular densities to assess the ability of blocking antibodies is a future challenge. Our preliminary experiments have shown that when MHC Class II and a costimulatory ligand CD80 are reconstituted into SLBs at very high densities, particularly outside of a physiological range, TCRs and CD28 homogenously translocate at a T cell-SLB interface, respectively. Therefore, in a case of some ligands expressed in a high level, it may be a physiological phenomenon that receptors uniformly gather at immune synapses to introduce their signaling.

We demonstrated in this paper that anti-hPD-L1 could more effectively block hPD-1-hPD-L1 binding than anti-hPD-1 at a minimal concentration of antibodies, if hPD-L1 is expressed on tumors alone. In contrast, when hPD-L2 is solely expressed on tumors, anti-hPD-1 is just used for the treatment, because no anti-hPD-L2 are currently approved for therapeutic use. If any anti-hPD-L2 bearing high efficacies for hPD-1-hPD-L2 blocking, such as 24 F.10C12, are developed and approved for therapeutic use in the future, we may obtain sufficient efficacies with a smaller dose of anti-hPD-L2 compared to anti-hPD-1. When both hPD-L1 and hPD-L2 are expressed on a tumor, we must choose anti-hPD-1 because of the current unavailability of commercial anti-hPD-L2. In these situations, higher doses of anti-hPD-1 might be required for therapy compared to anti-hPD-L1 and anti-hPD-L2 for hPD-L1- and hPD-

L2-expressing tumor, respectively. Thus, our evaluation strategy, including high-resolution imaging, is advantageous in that it can identify the characteristics of available ICIs so as to select the appropriate one for each patient and may also be useful to compare the efficacies of new drugs with others used in the drug development process.

In recent years, several combination therapies of antibodies have been shown to have efficacies in various diseases or cancer types[39–42]. This evidence is based on the idea that a synergistic effect can be expected from combining several antibodies against different targets or that, even if one antibody loses binding capacities to the target molecule due to mutation, the inhibitory effect of the other antibody will ensure a certain antitumor effect. Similar to ICIs, the recent COVID-19 pandemic coincidentally demonstrated some predominancies of a cocktail of antibodies with various epitopes[41,42]. The most popular combination of ICIs may be the cooperation of nivolumab and anti-hCTLA-4, ipilimumab, which has been reported to have efficiencies and acceptable side effects[39,40]. Interestingly, the combination of pembrolizumab, anti-hPD-1, and ipilimumab, anti-hCTLA-4, was less effective while those of nivolumab, anti-PD-1, plus ipilimumab, anti-hCTLA-4, was effective if the patients were inoculated with such combinations of ICIs with similar dosages[43]. This inconsistency in clinical output between these two combination therapies might be explained by the difference in epitopes to which anti-hPD-1 binds, but the precise mechanisms remain elusive. Our experiments demonstrated that pembrolizumab could inhibit hPD-1-hPD-L1 binding at a relatively lower concentration and that nivolumab showed similar effects at a more than tenfold higher concentration of antibodies. When just nivolumab was administered in small doses, it may not have been able to totally inhibit PD-1-PD-L1 binding; therefore, the addition of ipilimumab may have resulted in compensating antitumor effects that are synergistically collaborating with nivolumab.

In this report, we compared the blocking effects of PD-1-PD-L1 binding through the combinational use of extremely low doses of anti-hPD-1 and anti-hPD-L1 to those from a single use of each antibody at a relatively higher dose, and suggested that this combinational use will be more effective than a single use at a sufficient dose. When both hPD-L1 and hPD-L2 are expressed, it may be useful to use multiple types of antibodies with different targets in combination at low concentrations. In the case of a tumor detected by both hPD-L1 and hPD-L2, it will be useful to select multiple clones against different targets and to use them at low concentrations for the blocking of both hPD-1-hPD-L1 and -hPD-L2 binding. This will also lead to savings in medical resources and costs if the combinational use of various ICIs in a smaller amount than currently used could be expected to have a greater effect on tumor immunotherapy. Since tumors are continuously inducing genetic mutations, we occasionally find unexpected mutations in hPD-1. The multiple use of various clones would minimize the undesirable reduction in the antitumor effects by ICIs introduced by these mutations.

In this paper, we evaluated the blocking efficiencies of various anti-hPD-1, anti-hPD-L1, or anti-hPD-L2 against hPD-1-hPD-1 ligand bindings and also their combination by biochemical, physiological, and molecular imaging techniques. To choose the optimal therapies for individual cancer patients, the basic experiments of PD-1, including the molecular imaging in this paper, will become increasingly important and these kinds of studies will be applied to clinical practice in the future.

## Methods
### Ethical statement
This research complies with all relevant ethical regulations. All experiments in this study were approved by the Ethics Committee of Tokyo Medical University (TS2021-0527, T2022-0023).

### Reagents
The antibodies and reagents were purchased as follows: anti-IL-2 (1:500, JES6-1A12, e-Bioscience, 14-7022-85, RRID:AB_468406), biotin-labeled anti-IL-2 (1:1000, JES6-5H4, e-Bioscience, 13-7021-85, RRID:AB_466899), PE–anti-I-A/I-E (2.5 µg/ml, M5/114.15.2, BioLegend, 107608, RRID:AB_313323), PE–anti-mPD-1 (2.5 µg/ml, 29 F.1A12, BioLegend, 135206, RRID:AB_1877231), anti-mPD-1 (2.5 µg/ml, 29F.1A12, BioLegend, 135248, RRID:AB_2783091), anti-hPD-L1 (2.5 µg/ml, 29E.2A3, BioLegend, 329746, RRID:AB_2783199), and anti-hPD-L2 (2.5 µg/ml, 24F.10C12, BioLegend, 329624, RRID:AB_2819957), PE–anti-hPD-1 (10 µl, MIH4, BD bioscience, 557946, RRID:AB_647199), PE–anti-hPD-L1 (10 µl, MIH1, BD bioscience, 557924, RRID:AB_647198), PE–anti-hPD-L2 (10 µl, MIH18, BD bioscience, 558066, RRID: AB_647197), Alexa Fluor 647–labeled anti-pCD3ζ (2.5 µg/ml, K25-407.69, BD bioscience, 558489, RRID: AB_647152), and Alexa Fluor 647–labeled anti-pSLP-76 (2.5 µg/ml, J141-668.36.58, BD bioscience, 558438, RRID:AB_647159), rabbit polyclonal anti-SHP1 (1:500, C-19, Santa Cruz Biotechnology Inc., sc-287, RRID:AB_2173829) and mouse anti-SHP2 (1:1000, B-1, Santa Cruz Biotechnology Inc., sc-7384, RRID:AB_ 628252), anti-Erk (1:1000, Cell Signaling Technology, 4695S, RRID: AB_390779), anti-pErk (1:1000, Cell Signaling Technology, 4370S, RRID: AB_2315112), anti-PLCγ (1:1000, Cell Signaling Technology, 5690S, RRID: AB_10691383), anti-pPLCγ (1:1000, Cell Signaling Technology, 8713S, RRID: AB_10890863), anti-Akt (1:2000, Cell Signaling Technology, 4691S, RRID: AB_915783), anti-pAkt (1:1000, Cell Signaling Technology, 4060S, RRID: AB_2315049), and HRP-anti-rabbit IgG polyclonal Abs (1:10,000, Cell Signaling Technology, 7074S, RRID: AB_2099233), HRP-anti-mouse IgG polyclonal Abs (1:10,000, Cappel, 55550), pembrolizumab (MCE, HY-P9902), nivolumab (MCE, HY-P9903), durvalumab (MCE, HY-P9919), and atezolizumab (MCE, HY-P9904), APC-anti-human IgG (H + L) (2.5 µg/ml, Jackson Immuno Research, 705-136-147, RRID: AB_2340407), DyLight 650 and 549 labeling kits (Thermo Fisher Scientific, 84535 and 53044), HaloTag (HT) STELLA Fluor 650 and TMR ligands (Promega, GCKA308-01 and G8252), SNAP-Cell 647-Sir (New England BioLabs, S9102S), MCC$_{88-103}$ (ANERADLIAYLKQATK, GenScript), and OVA$_{257-264}$ (SIINFEKL, GenScript) peptides. A B cell hybridoma producing anti-CD28 (PV-1) was provided by R. Abe (Tokyo University of Science, Noda, Japan); anti-CD3ζ (145-2c11) by J. Bluestone (University of California, San Francisco, USA); anti-TCRβ (H57-597) by R. T. Kubo (Cytel Corp., CA, USA); and anti-I-E$^k$ (14-4-4) and anti-ICAM-1 (YN1/1.7.4) by M. L. Dustin (University of Oxford, UK); anti-hPD-L1 (MIH1)[44] and anti-hPD-L2 (MIH18)[44,45] by M. Azuma (Tokyo Medical and Dental University, Tokyo, Japan).

### Mice and cells
All animal experiments were performed in accordance with a protocol approved by the Animal Care and Use Committee of Tokyo Medical University (H30-0044, H31-0065, R2-0001). AND-TCR-Tg mice were provided by Dr. S. M. Hedrick (University of California San Diego, San Diego, CA); $Rag2^{-/-}$ mice by Dr. F. Alt (Boston Children's Hospital, Boston, MA); OT-I TCR-Tg $Rag2^{-/-}$ mice by Dr. W. Health (University of Melbourne, Melbourne, Australia); and $Pdcd1^{-/-}$ mice from RIKEN BRC. Mice were maintained in specific pathogen-free conditions with a 12 h light/dark cycle at 22 °C and controlled humidity (60 ± 10%) at Tokyo Medical University. All experiments were performed on 6-10 weeks old mice, age- and gender-matched. Experimental and control animals were co-housed. Mice were humanely euthanized by cervical dislocation once they reached the endpoints, like reaching 2000 mm$^3$ in tumor volume or loss of weight/mobility/body condition and severe neurological disabilities.

The DC-1 fibroblast cell expressing I-E$^k$ and ICAM-1 was provided by J. Kaye (Cedars-Sinai Medical Center, Los Angeles, CA). PLAT-E, the retrovirus packaging cell line, was provided by G. Nolan (Stanford University, Stanford, CA). Human lung cancer cell lines H460, H1299, and HCC827 were purchased from ATCC (ATCC, NCI-H460,

RRID:CVCL_0459; ATCC, NCI-H1299, RRID:CVCL_0060; ATCC, HCC827 PFR1, RRID:CVCL_DH92). BHK, EL-4, and E. G7-OVA Cell Line were purchased from ATCC (ATCC, ACC-61, RRID:CVCL_1915; ATCC, TIB-39, RRID:CVCL_0255; ATCC, CRL-2113, RRID:CVCL_3505). The T cell hybridoma expressing the AND-TCR (AND-TCR T cell hybridoma, 2D12) was established by cell fusion of activated AND-TCR-Tg CD4+ T cells with lymphoma cell line, BW5147[46]. We completely deleted mPD-1 (guides: CACCGATAAGATCCTCCGACCAGT, AAA-CACTGGTCGGAGGATCTTATC) on EL-4, E.G7, and 2D12 cells by CRISPR-Cas9 system (PX458, addgene, http://n2t.net/addgene:48138, RRID:Addgene_48138).

## Plasmid construction

EGFP-tagged hPD-1 (forward primer: CCGGAATTCGCCACCATGCA-GATCCCACAGGCGCC, reverse primer: CCGCTCGAGGAGGGGCCAA-GAGCAGTGTC), mSHP1 (forward primer: CCGGAATTCGCCACCATGT TGTCCCGCGGGTGGTT, reverse primer: CCGCTCGAGCTTCCTCTT-GAGAGAACCTT), and mSHP2 (forward primer: CCGGAATTCGCCAC-CATGACATCGCGGAGATGGTT, reverse primer: CCGGTCGACTCTG AAACTCCTCTGCTGCT) were generated by polymerase chain reaction (PCR) and subcloned into retroviral vector, pMXs and/or pMCs (kindly provided by Dr. T. Kitamura, University of Tokyo, Japan)[47]. SNAP-tag (New England BioLabs, N9183), HaloTag (Promega, G9651)-tagged hPD-1, hPD-L1 (forward primer: CCGGAATTCGCCACCATGAGGA-TATTTGCTGTCTTTA, reverse primer: CCGCTCGAGCGTCTCCTC-CAAATGTGTATC), or hPD-L2 (forward primer: CCGGAATTCGCCAC CATGATCTTCCTCCTGCTAATG, reverse primer: CCGCTCGAGGA-TAGCACTGTTCACTTCCC) and Renilla luciferase (RLuc) 8 (forward primer: CGGGAATTCGCCACCATGGCTTCCAAGGTGTACGACCCCGA, reverse primer: CTGGCGGCCGCTTACTGCTCGTTCTTCAGCACTCT) were generated by PCR and subcloned into the pMXs retroviral vector. A healthy donor consented to donate peripheral blood as a source of cDNA for subcloning of the human molecules in this paper. pMXs-RLuc8 was constructed by PCR using Yellow Nano-lanterns (kindly provided by Dr. Y. Okada, Riken, Japan)[48] as a template.

## Primary cell culture and transduction

A packaging cell, PLAT-E (provided by G. Nolan, Stanford University, Stanford, CA) transiently transduced with retroviral vectors using Lipofectamine 2000 (Invitrogen, 11668019). The supernatants were concentrated 40 to 80-fold by centrifugation at 8,000 $g$ for 12 h. AND-TCR-Tg CD4+ T cells were purified from AND-TCR-Tg $Pdcd1^{-/-}$ $Rag2^{-/-}$ mice and stimulated with 5 μM MCC$_{88-103}$ and irradiated spleen cells from B10.BR mice. OT-I TCR-Tg CD8+ T cells were purified from OT-I TCR-Tg $Pdcd1^{-/-}$ $Rag2^{-/-}$ mice and stimulated with 100 nM OVA$_{257-264}$, 200 U/ml recombinant mouse IL-2 (Peprotech, 212-12), and irradiated spleen cells from B6 mice. One day after stimulation, the cells were suspended in retroviral supernatant with 10 μg/ml polybrene (Sigma-Aldrich, TR-1003) and 200 U/ml mouse IL-2 and centrifuged at 1000 × $g$ for 90 min at 37 °C. On day 2, the cells were sorted to obtain the populations with homogeneous fluorescence intensity and were then maintained in RPMI1640 medium (Sigma-Aldrich, R6504-10L) containing 10% FCS (Thermo Fisher Scientific, 10270106) and mouse IL-2.

## Microscopy

Cells expressing the proteins tagged with GFP and/or HaloTag stained by fluorescent-labeled H57 Fab and/or TMR-(Promega, G8252) or Stella650-labeled HaloTag ligand (Promega, GCKA308-01) were allowed to settle onto an SLB. The phosphorylation of CD3ζ and SLP-76 were detected using fluorescent-labeled anti-pCD3ζ and pSLP-76, respectively. Images were acquired using a confocal laser scanning microscope (TCS SP8, Leica Microsystems) comprising a 63× oil-immersion objective lens, high sensitivity HyD detectors, and 488, 561, and 633 nm laser lines. LAS X software (Leica, Germany) was utilized for image acquisition. A TIRF analysis system was set up on a

conventional inverted microscope (Ti-LAPP, Nikon, Tokyo, Japan) outfitted with a TIRF objective lens (Nikon), a scientific CMOS camera (ORCA flash 4.0, Hamamatsu photonics) and fiber-coupled 488 nm lasers. The exposure time was set at 100 ms with 2.5 s-interval between time points. NIS-elements software (Nikon) was used for image acquisition. ImageJ software (NIH, Bethesda, MD, USA, RRID:SCR_003070) was used for image processing and final figure preparation.

## Glass-supported lipid planer bilayers

The purification and fluorescent labeling of GPI-anchored proteins have been established according to the protocols[23]. The mouse MHC class II molecule I-E$^k$ with a GPI anchor (I-E$^k$−GPI), the mouse MHC class I molecule H-2K$^b$ with a GPI anchor (H-2K$^b$−GPI) and mouse ICAM-1 with a GPI anchor (ICAM-1−GPI) were purified from transfected Chinese hamster ovary and baby hamster kidney cells, respectively, and were incorporated into dioleoyl phosphatidylcholine liposomes (Avanti Polar Lipids). BHK cells (ATCC, ACC-61, RRID:CVCL_1915) highly expressing hPD-L1−GPI or hPD-L2−GPI were established. hPD-L1−GPI and hPD-L2−GPI were purified from the lysates by affinity column with 29E.2A3 (BioLegend, 329746, RRID:AB_2783199, anti-hPD-L1) and MIH18 (provided by M. Azuma, anti-hPD-L2)[44,45], respectively. The expression level of each GPI-anchored protein on the planar bilayer was quantified using silica beads with a diameter of 5 μm (Bangs Laboratories, SS05N)[27]. The densities were calculated based on the standard beads, Quantum FITC-5 MESF (Bangs Laboratories, 555p), and adjusted to the approximate concentration by comparison with natural APCs: I-E$^k$, 200 molecules/μm$^2$; H-2K$^b$, 200 molecules/μm$^2$; mICAM-1, 150/μm$^2$; hPD-L1, 17.25–600/μm$^2$, and hPD-L2 200/μm$^2$. We prepared planar bilayers by mixing GPI-anchored proteins, dropping them on clean glass (40 mm glass coverslips, Biotechs), and overlaying with a clean cover glass (Fisherbrand, Circles; Size: 12 mm) for 30 min. The planar bilayers were loaded with 10 μM MCC$_{88-103}$ (GenScript) or 10 μM OVA$_{257-264}$ in citrate buffer, pH 4.5, for 24 h at 37 °C, blocked with 5% nonfat dried milk (Cell Signaling Technology, 9999S) in PBS for 30 min at 37 °C, removed the cover glass, and left to stand in the assay medium (Hepes-buffered saline, Sigma-Aldrich, H3375-250G) containing 1% FCS (Thermo Fisher Scientific, 10270106), 2 mM MgCl$_2$, and 1 mM CaCl$_2$ in a flow cell chamber system (Bioptechs).

## Image processing

The size and fluorescence intensity of each region were analyzed in all images by ImageJ. The fluorescence intensities were quantified based on the raw imaging data using the following formula. [intensity of fluorescence at each spot on a diagram] − [minimal intensity of each fluorescence on the entire line])/([mean intensity of each fluorescence on the entire line] − [minimal intensity of each fluorescence along the entire line][27]. Pearson's colocalization coefficients (PCCs) were subsequently calculated from each fold intensities.

10 TCR microclusters per cell were randomly selected from 10 (Fig. 1d) or 30 (Fig. 5b and Supplementary Fig. 1g) cells, and the percentages of colocalization between TCR microclusters and hPD-1 microclusters were calculated and presented as TCR microclusters colocalized with hPD-1 (%).

The percentages of cells forming more than three hPD-1 microclusters are presented as % of T cells forming hPD-1 microclusters.

## T cell−APC conjugation assay

DC-1 or DC-1 cells expressing hPD-L1-HaloTag or SNAP-tag and/or hPD-L2-HaloTag were prepulsed with 1 μM MCC$_{88-103}$ overnight at 37 °C and prestained with SNAP-Cell 647-Sir (New England BioLabs, S9102S, red) and/or HaloTag ligand-TMR (cyan). mPD-1-deleted 2D12 cells, the T cell hybridoma expressing the AND-TCR, expressing hPD-1-EGFP were cultured with DC-1 cells with or without 0.1 or 10 μg/ml anti-hPD-1 and/or anti-hPD-L1 and/or anti-hPD-L2 antibody. mPD-1-deleted 2D12 cells

expressing hPD-1-HaloTag and EGFP-mSHP1 or -mSHP2 were cultured with DC-1 cells. The conjugates were visualized by confocal microscopy.

## Immunoprecipitation and Western blotting

DC-1 cells were prepulsed with 5 μM $MCC_{88-103}$ overnight at 37 °C and washed before the assay. $2 \times 10^6$ mPD-1-deleted 2D12 cells transduced with hPD-1 were stimulated with $2 \times 10^6$ DC-1 cells not transduced or transduced with hPD-L1. The cells were lysed with the lysis buffer (50 mM Tris-HCl, 50 mM NaCl, and 5 mM EDTA) containing 1% NP-40. Whole cell lysates (WCLs) or those immunoprecipitated by anti-GFP (MBL International, D153-11, RRID:AB_2893312) were blotted with anti-GFP (1:5000, Miltenyi Biotec, 130-091-833, RRID:AB_247003), anti-mSHP1 (1:500, Santa Cruz Biotechnology, sc-287, RRID:AB_2173829), anti-mSHP2 (1:1000, Santa Cruz Biotechnology Inc., sc-7384, RRID:AB_628252), anti-PLCγ (1:1000, Cell Signaling Technology, 5690, RRID:AB_10691383), anti-pPLCγ (1:1000, Cell Signaling Technology, 8713, RRID:AB_10890863), anti-Akt (1:2000, Cell Signaling Technology, 4691, RRID:AB_915783), anti-pAkt (1:1000, Cell Signaling Technology, 4060, RRID:AB_2315049), anti-Erk (1:1000, Cell Signaling Technology, 4695, RRID:AB_390779), or anti-pErk (1:1000, Cell Signaling Technology, 4370, RRID:AB_2315112) as a first antibody and HRP-anti-rabbit IgG polyclonal Abs (1:10,000, Cell Signaling Technology, 7074, RRID:AB_2099233) as a second one. Each intensity of band was calculated by ImageJ (RRID:SCR_003070).

## Flow cytometry

Staining antibodies were used at a concentration of 0.01–100 μg/ml. A cell sorter, SH800S (Sony), was used for cell isolation and cell analyzers, FACS Canto II (BD, 07B1X00003000102) and Guava easyCyte (MERCK, 0500-5007JPK) were used for analysis. Data were depicted using FlowJo (RRID:SCR_008520).

## T cell stimulation assay

$2 \times 10^4$ mPD-1-deleted 2D12 cells or $1 \times 10^5$ $Pdcd1^{-/-}$ AND-TCR-Tg $CD4^+$ T cells were stimulated with $2 \times 10^4$ DC-1 cells expressing hPD-L1 and/or hPD-L2 with 1 μM $MCC_{88-103}$ in the presence or absence of anti-hPD-1 and/or anti-hPD-L1 and/or anti-hPD-L2 antibody. The concentration of IL-2 was measured by ELISA 16 h after stimulation. All experiments were performed in triplicate.

## CTL killing assay

RLuc8-introduced and mPD-1-deleted EL-4 cells (ATCC, TIB-39, RRID:CVCL_0255) were used as a target cell. At the indicated E/T ratios, hPD-1-transduced $Pdcd1^{-/-}$ OT-I TCR-Tg $CD8^+$ T cells were cocultured with 1 nM $OVA_{257-264}$ pulsed $1 \times 10^5$ EL-4 cells expressing hPD-L1 for 16 h in the presence or absence of anti-hPD-1 or anti-hPD-L1 antibodies. After treatment with coelenterazine, RLuc8 substrate (FUJIFILM Wako, 031-22993), the intensity of RLuc8 luminescence in live target cells was measured using a lumino image analyzer, ImageQuant LAS4000 mini (GE Healthcare). All experiments were performed in triplicate.

## In vivo tumorigenicity assay

$1 \times 10^6$ mPD-1-deleted E. G7-OVA Cell Line (ATCC, CRL-2113, RRID:CVCL_3505) reconstituted by hPD-L1 were subcutaneously inoculated in 100 μl PBS in the dorsal region of $Rag2^{-/-}$ mice. Tumors were allowed to grow for 8 to 10 days before treatments (tumor area between 90 and 400 $mm^2$). Tumor size was blindly measured using calipers. Mice received 100 μl PBS containing $1.5 \times 10^6$ activated $Pdcd1^{-/-}$ OT-I TCR-Tg $CD8^+$ T cells expressing hPD-1 by intravenous injection in the tail vein. Two days later, mice were injected intraperitoneally with nivolumab (MCE, HY-P9903, RRID:AB_2810223) or durvalumab (MCE, HY-P9919) at 0.001 mg/kg or 0.1 mg/kg four times every 2–3 days.

## Quantification of microcluster formation

The ratio of the total area of microclusters to the area of the T cell-SLB interface in individual T cells was quantified by using available plugins in Fiji as mentioned in Supplementary Fig. 5a, b.

## Statistics and reproducibility

Data were presented as the mean ± standard deviation (SD). Statistical analysis was performed by the Student's $t$-test or one-way analysis of variance (ANOVA) using GraphPad Prism (RRID:SCR_002798). $p$ values < 0.05 were considered to be statistically significant. Reproducibility, including biological independent sample sizes and replicates, are stated in each figure legend.

## Reporting summary

Further information on research design is available in the Nature Portfolio Reporting Summary linked to this article.

## Data availability

All data supporting the conclusions included in the manuscript are available within the paper and its supplementary information. Source data are provided with this paper.

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

## Acknowledgements

We thank Dr. Toshio Kitamura for pMXs and pMCs retroviral vector, and Mai Kozuka for secretarial assistance. This work was supported by JSPS KAKENHI (JP25113725, JP15H01194, JP16H06501, JP17H03600, JP19K22545, JP20H03536, JP23H02775, JP23H04790, T.Y), PRESTO (U1114011, T.Y.) from Japan Science and Technology Agency, the Takeda Science Foundation (T.Y.), and the Naito Foundation (4465-135, T.Y.).

## Author contributions

W.N., E.W., and T.Y. designed the research; W.N., E.W., H.M., K.S., Y.Y., R.M., T.N., T.T., H.T., M.F., H.N., A.T., and T.Y. performed the research; M.A. and T.Y. contributed the new reagents; W.N. and H.M. analyzed the data; E.W., A.T., M.S., and T.Y. supervised the research; and W.N. and T.Y. wrote the paper.

## Competing interests

The authors declare no competing interests.
