## [Peer Review File · Nature Communications]

Evaluation of therapeutic PD-1 antibodies by an advanced single-molecule imaging system detecting human PD-1 microclustersEditorial Note: Parts of this Peer Review File have been redacted as indicated to remove third-party material where no permission to publish could be obtained.

REVIEWER COMMENTS

Reviewer #1 (Remarks to the Author):

In this study, the authors developed an advanced single-molecule imaging system for visualizing human PD-1 microclusters to apply to the evaluation of various therapeutic PD-1 and PD-L1 antibodies. This group already has reported several papers on the role of mouse PD-1/PD-L1 and PD-1/PD-L2 interaction in T-cell responses using the Supported Lipid Bilayer (SLB) and TIRF microscopy. Thus, their experimental system seems to be very reliable and data of high quality was presented. However, overall, a majority of data is related to confirmatory findings (especially, Fig 1 to 3) as the authors mentioned several times in the manuscript. This is a case where human PD-1/PD-L1 is applied to an already established in vitro mouse PD-1/PD-L1 system. Thus, it is questionable whether this study fits the scope of this journal.

Additional concerns include that the evidence of the advantage of this advanced single-molecule imaging system in the evaluation of therapeutic PD-1 antibodies is not clear. I suggest that the author should provide more data where their imaging system is superior to conventional functional assays, such as IL-2 production and CTL lysis. Considering that it is applied to evaluate the efficacy of hPD-1/hPD-1 ligands antibody treatments in cancer patients, it would be good to evaluate their findings using some patient samples as well as in vitro imaging systems.

In fact, the interesting data is probably PD-1-mediated T cell suppression presented in Figures 6 and 7, where authors showed PD-1-mediated T cell suppression more effectively with combinational use of anti-hPD-1 and anti-hPD-1 ligands. If the mechanism underlying this finding can be elucidated by their imaging and biochemical assay system, then this manuscript has strength in terms of conceptual advance and novelty.

Reviewer #2 (Remarks to the Author):

Summary:

In this study, Nishi and co-workers revisited the in vitro and cellular reconstruction of human PD-1 signalosome in a similar manner to their recently published data [PMID: 33990697]. Here, they substituted the mouse PD-1 and PD-1 ligands with their human orthologs. They thoroughly validated the functionality of the reconstructed system by using cell-based, biochemical, and biophysical assays and presented very clearly the correlation between hPD-1 mediated microcluster formation and the disruption of hPD-1 signaling in mouse T cells. The main aim of the study was to develop a digital platform based on their in vitro reconstructed PD-1 signalosome for evaluating the blocking efficiency and effective concentration of therapeutic PD-1 inhibitors in an individual or combinatorial fashion. By using the reconstructed hPD-1 cellular system, the authors were capable of reporting valuable quantitative information in terms of dose-response curve and half maximum effective concentrations of ICIs in clinical use. In addition, they provide evidence for the synergistic effect of PD-1 and PD-L1 blockers.

The work is very well done and organized. Data based on the functional assays is convincing and correlate to a large extent with previous findings. Their imaging system provides an excellent visual aid to evaluate the blocking efficiency of the tested ICIs, however, it fell short to provide a digital readout and cannot be used solo as a predictive tool.

Significance:

PD-1 blockers have revolutionized cancer therapy and their number is growing constantly. However, still there is limited information comparing the efficiency of different ICIs and their combinatorial use. The development of methods for thorough characterization and comparison between different PD-1 inhibitors can pinpoint their optimal use in clinical settings. Currently, the main approaches are based on functional assays, this manuscript has the potential to promote the development of a novel alternative for rapid, cost-effective, and combinatorial drug screening. I think that, in general, this is an interesting study. However, it has some caveats and open issues that should be clarified before its publication.

Major:

1. PD-L1 is a valuable predictive biomarker for ICIs efficiency. One of the main benefits of SLB is the possibility of fine-tuning the concentration of molecules of interest in contrast to the standard functional assays. For the majority of experiments the authors are using 150 molecules/ μm^2 PD-L1 which fits in the physiological range (the exact range should be clearly stated in the main text). It is peculiar that the authors are not exploring the full capacity of their system and are not testing the PD-1 blockers on substrates with different PD-L1 densities. In addition, they are not providing an explanation regarding the abrogated PD-1 microcluster formation when using high PD-L1 density as shown in Supplementary Fig. 1a. It is necessary to discuss further this observation and the potential limitation of the system.

2. The major gap in this study is the lack of digital readout from the imaging system. Although the study is built around it the important findings are based on the cell-based systems and the in vitro reconstruction assay serves as an accompanying visual aid. The manuscript is missing a quantitative method to analyze the image data and correlate the computed results with the performed functional assays. The development of such a method will significantly increase the impact of the study. Is there any difference in the number of PD-1 microclusters per cell when titrating the inhibitor concentration (Fig. 4a)?

3. It is somewhat pity that the authors did not use the imaging system to show the combinatorial effect of PD-1 blockers (Fig. 7), the authors need to provide this data. In addition, they did not use it when testing the PD-1 drugs inhibitory effect with OT-I TCR Tg cells. It would have been quite easy for them to place the labeled hPD-1 and perform the in vitro experiment together with the accompanying cellular functional assay. The IL-2 assay used to derive the E50 values is also missing, it would be nice to be performed in order to show the compatibility of their PD-1 reconstructed system in different T cell types.

4. The calculated EC50 values are 100 folds higher than the published EC50 for the same inhibitors estimated by performing a functional assay using human cells [PMID: 31391510]. The total number of PD-1 and PD-1 ligands expressed by the used cell types need to be included in order to enable adequate comparison between different studies.

Minor:

5. It is not clear how the colocalization of TCR and PD1 microclusters is computed. The authors should provide more details in the method section. In addition, in Supplementary Fig. 1c the y-axis label differs from the legend description 'percentages of the T cells forming hPD-1 microclusters'. The readers can benefit from increasing the number of ticks on the y-axis for this type of graph.

6. Although the authors have demonstrated clearly the binding between the PD-1 and SHP2 as well as the biological outcome of PD1 microcluster formation they should be careful in disregarding the possibility of SHP1 involvement especially since the interaction is mediated between hPD-1 and mouse SHP1/2.

7. The ratio computed based on the Western blot results shown in Fig 2e will be better presented in a graph.

8. Pearson's correlation coefficient (PCC) abbreviation should be included in the legend or in the text (see Fig. 2d and Supplementary Fig. 2c).

9. There is a lack of information regarding the choice of inhibitors' concentration used in the first inhibitory assay presented in Fig. 3. In addition, the presented images in Fig. 3a are not correlating well to the computed data shown in Fig. 3b.

10. Since the authors have already performed the flow cytometry based binding assay as shown in Supplementary Fig. 4 it will be of interest to calculate and summarize the EC50 values and the 95% CI and compare them to the functional assays derived estimates.

11. It would be nice if the authors place in the legend which cell type was used as effector and which one as a target (Fig. 4e).

12. The statement in line 287 is quite inaccurate 'As shown in Table 1, the EC50s calculated by inhibitory efficiencies on IL-2 production from these T cells were actually lower compared to EC90 and EC98 for all antibodies. However, the combination of anti-hPD-1 and anti-hPD-L1 at their respective EC50s increased the IL-2 production more than a single antibody at EC90 (Fig. 7a and b).'

13. The PD-L1 in Figure 5e should be substituted with PD-L2

14. For consistency I would like to see the % T cell forming PD-1 microclusters upon inhibitors application based on the images shown in Supplementary Fig. 4a

15. The authors should provide information how the increased ratio of IL-2 in Fig. 7b and statistical analysis of the data presented in Fig. 7c

16. There is a complete lack of information regarding the formation of SLB a central piece of the imaging system. It will be of interest to know the exact procedure, the type of substrate used as well as the possibility of using high-throughput formats.
17. In general, throughout the manual there is a lack of citations.

The following are the point-by-point response to the reviewers' comments.

Reviewer #1 (Remarks to the Author):

In this study, the authors developed an advanced single-molecule imaging system for visualizing human PD-1 microclusters to apply to the evaluation of various therapeutic PD-1 and PD-L1 antibodies. This group already has reported several papers on the role of mouse PD-1/PD-L1 and PD-1/PD-L2 interaction in T-cell responses using the Supported Lipid Bilayer (SLB) and TIRF microscopy. Thus, their experimental system seems to be very reliable and data of high quality was presented. However, overall, a majority of data is related to confirmatory findings (especially, Fig 1 to 3) as the authors mentioned several times in the manuscript. This is a case where human PD-1/PD-L1 is applied to an already established in vitro mouse PD-1/PD-L1 system. Thus, it is questionable whether this study fits the scope of this journal.

Additional concerns include that the evidence of the advantage of this advanced single-molecule imaging system in the evaluation of therapeutic PD-1 antibodies is not clear. I suggest that the author should provide more data where their imaging system is superior to conventional functional assays, such as IL-2 production and CTL lysis.

Thank you for your meaningful suggestions. We agree with you that the conventional functional assays can evaluate the ability of ICIs by itself. The strong point of our imaging system is that we can precisely determine the level of PD-1 ligands and evaluate the blocking effects of antibodies with various expression levels of the ligands widely mimicking APCs or cancer cells. We think that there is a benefit to being able to visually assess the abilities of ICIs as possible as a functional assay under more stable and changeable *in vitro* conditions with less biomaterials.

We have developed a new macro which can calculate the ratio of the total area of microclusters to the area of the T cell-SLB interface in individual T cells from imaging data. As the concentration of ICIs added on the SLBs increased, the areas were reduced and showed a strong negative correlation with IL-2 production in the functional assay, as we expected.

In the future, we hope that such digital readouts from imaging data will assist us to predict the efficacy of ICIs in patients prior to administration. We added the new data as Fig. 4b and d and Supplementary Fig. 5a and b and described it in page 10 line 235-238 and page 11 line 242-245 in the main text.

Considering that it is applied to evaluate the efficacy of hPD-1/hPD-1 ligands antibody treatments in cancer patients, it would be good to evaluate their findings using some patient samples as well as in vitro imaging systems.

We appreciate the reviewer's comment on this point and also recognize the importance of using the patient samples. Since it was quite tough to take human cancer samples into our in vitro experiments, we tested several cancer cell strains and choose a human lung cancer cell line, HCC827, as a target cell for both functional assays and cell imaging analysis. As expected, we got similar results with the experiments using murine cells, DC-1 or EL-4 cells. We added the new data as Supplementary Fig. 4h, i and j and described it in page 10 line 217-223 in the main text.

In fact, the interesting data is probably PD-1-mediated T cell suppression presented in Figures 6 and 7, where authors showed PD-1-mediated T cell suppression more effectively with combinational use of anti-hPD-1 and anti-hPD-1 ligands. If the mechanism underlying this finding can be elucidated by their imaging and biochemical assay system, then this manuscript has strength in terms of conceptual advance and novelty.

We understand and totally agree with the reviewer's comments. As the reviewer mentioned, it is an important future plan to elucidate deeply the mechanism underlying the efficacies of combinational use of antibodies. We investigated whether the efficacy of anti-PD-1 and anti-PD-L1 in combination could be detected by our imaging system. As expected, such combination was also effective in suppressing hPD-1 microcluster formation, consistent with the results of the functional analysis. We added the new data as Fig. 7c, d and e and explained it in page 14 line 325-334 in the main text.

Reviewer #2 (Remarks to the Author):

Summary:

In this study, Nishi and co-workers revisited the in vitro and cellular reconstruction of human PD-1 signalosome in a similar manner to their recently published data [PMID: 33990697]. Here, they substituted the mouse PD-1 and PD-1 ligands with their human orthologs. They thoroughly validated the functionality of the reconstructed system by using cell-based, biochemical, and biophysical assays and presented very clearly the correlation between hPD-1 mediated microcluster

formation and the disruption of hPD-1 signaling in mouse T cells. The main aim of the study was to develop a digital platform based on their in vitro reconstructed PD-1 signalosome for evaluating the blocking efficiency and effective concentration of therapeutic PD-1 inhibitors in an individual or combinatorial fashion. By using the reconstructed hPD-1 cellular system, the authors were capable of reporting valuable quantitative information in terms of dose-response curve and half maximum effective concentrations of ICIs in clinical use. In addition, they provide evidence for the synergistic effect of PD-1 and PD-L1 blockers.

The work is very well done and organized. Data based on the functional assays is convincing and correlate to a large extent with previous findings. Their imaging system provides an excellent visual aid to evaluate the blocking efficiency of the tested ICIs, however, it fell short to provide a digital readout and cannot be used solo as a predictive tool.

Significance:

PD-1 blockers have revolutionized cancer therapy and their number is growing constantly. However, still there is limited information comparing the efficiency of different ICIs and their combinatorial use. The development of methods for thorough characterization and comparison between different PD-1 inhibitors can pinpoint their optimal use in clinical settings. Currently, the main approaches are based on functional assays, this manuscript has the potential to promote the development of a novel alternative for rapid, cost-effective, and combinatorial drug screening.

I think that, in general, this is an interesting study. However, it has some caveats and open issues that should be clarified before its publication.

Major:

1. PD-L1 is a valuable predictive biomarker for ICIs efficiency. One of the main benefits of SLB is the possibility of fine-tuning the concentration of molecules of interest in contrast to the standard functional assays. For the majority of experiments the authors are using 150 molecules/ μm^2 PD-L1 which fits in the physiological range (the exact range should be clearly stated in the main text). It is peculiar that the authors are not exploring the full capacity of their system and are not testing the PD-1 blockers on substrates with different PD-L1 densities. In addition, they are not providing an explanation regarding the abrogated PD-1 microcluster formation when using high PD-L1 density as shown in Supplementary Fig. 1a. It is necessary to discuss further this observation and the potential limitation of the system.

First, we apologize for the inadequate explanation of Supplementary Fig. 1a. The formation of hPD-1 microclusters cannot be clearly imaged by hPD-L1 at densities above 300 molecules/ μm^2 or less than 18.75 molecules/ μm^2 on SLBs. When there are extremely many or few receptors expressed on cells or ligands reconstituted onto SLBs, visualization of microclusters often does not work well. We described it in page 19 line 440-453 in the main text. Although within the range of at least 37.5-150 molecules/ μm^2 of hPD-L1 on SLBs, we could confirm the formation of hPD-1 microclusters and the antibody-specific blocking effects by anti-PD-1 or anti-PD-L1. These data are shown in Supplementary Fig. 6a and b and provided in page 12 line 267-271 in the main text.

Since the expression levels of PD-L1 in human cancer samples are usually assessed by histological staining, we have not been able to find the physiological density of hPD-L1 that is exactly mentioned in previous reports. We coated silica beads with lipid-bilayer containing hPD-L1 at various molecular densities and compared the level of hPD-L1 on silica beads with those of several lung cancer cell lines. As expected, densities of hPD-L1 on those cell lines varied extensively. Based on “depmap”, the database profiling hundreds of cancer cell line models for genomic information and sensitivity to genetic and small molecule perturbations, hPD-L1 is shown to be expressed at various levels in other cell strains (https://depmap.org/portal/interactive/?filter=slice%2Fcontext%2FLung%2Flabel®ressionLine=false&associationTable=false&x=slice%2Fexpression%2F3874%2Fentity_id&y=&color=, please see below). These data indicate that the range of 37.5 to 150 molecules/ μm^2 of hPD-L1 density is not so far from the physiological range. In the previous reports by Dustin Lab showing mPD-1 microclusters by using similar system, they reconstituted mPD-L1 at 275 molecules/ μm^2 on SLBs (Zinselmeyer, BH. *et al, J Exp Med.*, 2013). Because we think that the density of PD-L1 at 275 molecules/ μm^2 will be slightly higher than those of normal cells that we checked, therefore we set the density of hPD-L1 at 150 molecules/ μm^2 . We added the new data as Supplementary Fig. 1b and c and described it in page 6 line 122-129 in the main text.

[REDACTED]

Finally, we performed functional assays and cell-cell conjugation experiments by using HCC827 cell, which was one of the cell lines tested in above, and showed that the inhibitory effects of the hPD-1-hPD-L1 binding and the recovery cytokine production by anti-hPD-1 or anti-hPD-L1. We added the new data as Supplementary Fig. 4h, i, and j and explained it in page 10 line 217-223 in the main text.

2. The major gap in this study is the lack of digital readout from the imaging system. Although the study is built around it the important findings are based on the cell-based systems and the in vitro reconstruction assay serves as an accompanying visual aid. The manuscript is missing a quantitative method to analyze the image data and correlate the computed results with the performed functional assays. The development of such a method will significantly increase the impact of the study. Is there any difference in the number of PD-1 microclusters per cell when titrating the inhibitor concentration (Fig. 4a)?

We really thank for the meaningful suggestions. To answer the reviewer's comments, we developed a new macro which can calculate the ratio of the total area of microclusters to the area of the T cell-SLB interface in individual T cells from imaging data. Because

the size of each microcluster varies from cluster to cluster, it was more informative to examine a correlation of T cell response with the total area of microclusters, not simply with the number of them. As shown in Fig. 4b, the total area of hPD-1 microclusters was reduced if the blocking antibody was added at progressively higher concentrations to all antibodies. The concentration required to reduce the area was also found to vary from antibody to antibody. The total area of hPD-1 microclusters showed a strong negative correlation with the results of functional analysis, indicating that the total area of microclusters calculated from imaging data may be used as a tool for predicting efficacy of antibodies. We added the new data as Fig. 4b and d and Supplementary Fig. 5a and b and explained it in page 10 line 235-238 and page 11 line 242-245 in the main text.

3. It is somewhat pity that the authors did not use the imaging system to show the combinatorial effect of PD-1 blockers (Fig. 7), the authors need to provide this data.

We understand and totally agree with the reviewer's comments. We examined the appropriate concentration of anti-PD-1 or anti-hPD-L1 for the new experiment and found that the combination use of anti-hPD-1 and anti-hPD-L1 in each EC₂₀ effectively inhibited the formation of hPD-1 microclusters. We finally show the combinational effect of PD-1 blockers in this imaging system, added the new data as Fig. 7c, d and e and described it in page 14 line 325-334 in the main text.

In addition, they did not use it when testing the PD-1 drugs inhibitory effect with OT-I TCR Tg cells. It would have been quite easy for them to place the labeled hPD-1 and perform the in vitro experiment together with the accompanying cellular functional assay. The IL-2 assay used to derive the E50 values is also missing, it would be nice to be performed in order to show the compatibility of their PD-1 reconstructed system in different T cell types.

To answer the reviewer's comment, we further examined the hPD-1 microcluster formation by using primary CD8⁺ T cells from OT-I TCR Tg cells and an SLB expressing hPD-L1 plus OT-I TCR-restricted MHC, H-2K^b. The results are same as those of AND TCR-Tg CD4⁺ T cells shown in Supplementary Fig. 4a and b. We added the new data as Supplementary Fig. 4f and g and described it in page 9 line 212-216 in the main text. The recovery of IFN γ production from CD8⁺ OT-I TCR Tg T cells by interfering hPD-1-hPD-L1 binding are depicted in Supplementary Fig. 5d and EC₅₀s of

inhibitory efficiencies on IFN γ production are in Supplementary Fig. 5e and Supplementary Table 2 and explained it in page 11 line 251-254 in the main text.

4. The calculated EC50 values are 100 folds higher than the published EC50 for the same inhibitors estimated by performing a functional assay using human cells [PMID: 31391510]. The total number of PD-1 and PD-1 ligands expressed by the used cell types need to be included in order to enable adequate comparison between different studies.

We totally agree the reviewer's comment for the importance of the number of hPD-1 and hPD-1 ligands expressed by the cells. We added flowcytometry data for each cell used in this paper in Supplementary Fig. 1c, 1d, 1e, 1h, 4d, 4e, 5f, and 7b. As the reviewer mentioned, the differences in EC₅₀s from the previous report (De Sousa Linhares, A. *et al*, *Sci. Rep.*, 2019) may be influenced by the expression levels of hPD-1 and hPD-1 ligands. Furthermore, in this paper, we used mouse-derived cells, but it may be better to use human cells to evaluate the blocking ability of therapeutic antibodies. This is our next research task. We discussed it in page 16 line 373-378 in the main text.

Minor:

5. It is not clear how the colocalization of TCR and PD1 microclusters is computed. The authors should provide more details in the method section. In addition, in Supplementary Fig. 1c the y-axis label differs from the legend description 'percentages of the T cells forming hPD-1 microclusters'. The readers can benefit from increasing the number of ticks on the y-axis for this type of graph.

Thank you for the comment. We changed the legend of Supplementary Fig. 1c (new Supplementary Fig. 1g) to 'TCR microclusters colocalized with hPD-1 (%)'.

We added the following text to the Materials and Methods for calculating the colocalization efficiencies of TCR and PD-1 microclusters. "10 TCR microclusters per cell were randomly selected from 10 (Fig. 1d) or 30 (Fig. 5b and Supplementary Fig. 1g) cells, and the percentages of colocalization between TCR microclusters and hPD-1 microclusters were calculated and presented as 'TCR microclusters colocalized with hPD-1 (%)'."

6. Although the authors have demonstrated clearly the binding between the PD-1 and SHP2 as well as the biological outcome of PD1 microcluster formation they should be careful in disregarding the possibility of SHP1 involvement especially since the interaction is mediated between hPD-1 and mouse SHP1/2.

We understand and totally agree with the reviewer's comments. As shown previously, murine (m) SHP2 was recruited to mPD-1 upon the mPD-1–mPD-L1 binding, whereas the mPD-1–mSHP1 association remained at background level (Okazaki, T. *et al*, *Proc. Natl. Acad. Sci. USA.*, 2001; Yokosuka, T. *et al*, *J. Exp. Med.*, 2012). But we cannot completely disregard the possibility that mouse SHP1 also acts in some way as mentioned by the reviewer, we add the next sentences in the text, page 17 line 392-396.

“Previous reports have shown that PD-1 strongly recruits SHP2, but not SHP1, in both T cells and B cells^{14, 27, 32}. On the other hand, some reports discussed that PD-1 recruits both SHP-1 and SHP-2 in T-cells³³. Although we could not confirm the recruitment of mSHP1 to hPD-1 in our experiments, we could not deny a possibility that mSHP1 may also act in a similar way as mSHP2.”.

7. The ratio computed based on the Western blot results shown in Fig 2e will be better presented in a graph.

To follow the reviewer's recommendations, we added new graphs as Fig. 2f.

8. Pearson's correlation coefficient (PCC) abbreviation should be included in the legend or in the text (see Fig. 2d and Supplementary Fig. 2c).

We added the term to a figure legend in Fig. 2d.

9. There is a lack of information regarding the choice of inhibitors' concentration used in the first inhibitory assay presented in Fig. 3. In addition, the presented images in Fig. 3a are not correlating well to the computed data shown in Fig. 3b.

Because flowcytometric analysis showed that all antibodies were sufficient to bind to almost of all their targets at least 2 µg/ml as concentration (Supplementary Fig. 3a and b), we chose 10 µg/ml. We mentioned it in page 8-9 line 191-193 in the main text.

Fig. 3b demonstrate percentages of cells forming more than three hPD-1 microclusters out of 30 cells each in Fig. 3a. We add this explanation at the Materials and Methods.

10. Since the authors have already performed the flow cytometry based binding assay as shown in Supplementary Fig. 4 it will be of interest to calculate and summarize the EC50 values and the 95% CI and compare them to the functional assays derived estimates.

Thank for the meaningful comments and suggestions. As a previous report (De Sousa Linhares, A. *et al*, *Sci. Rep.*, 2019), half maximum effective concentrations (EC₅₀) were calculated from the MFI values in Supplementary Fig. 3a and b using 4-parameter logistic function (Supplementary Fig. 3c and Supplementary Table 1). The EC₅₀s of each antibody were almost same in our experiment. We think that the discrepancy between our results and those in the previous paper could be due to the differences of the expression levels of hPD-1 or hPD-L1 molecules. We added the new data as Supplementary Fig. 3c and Supplementary Table 1 and described it in page 9 line 193-197 in the main text.

11. It would be nice if the authors place in the legend which cell type was used as effector and which one as a target (Fig. 4e).

To follow the reviewer's recommendations, we added the information of cell types to the figure legend. (new Fig. 4g).

12. The statement in line 287 is quite inaccurate 'As shown in Table 1, the EC50s calculated by inhibitory efficiencies on IL-2 production from these T cells were actually lower compared to EC90 and EC98 for all antibodies. However, the combination of anti-hPD-1 and anti-hPD-L1 at their respective EC50s increased the IL-2 production more than a single antibody at EC90 (Fig. 7a and b).'

To answer the reviewer's comment, we rewrite the manuscript as follows in page 14 line 319-323 in the main text.

'As shown in Table 1, compared to EC₉₀ and EC₉₈, the EC₅₀ required to restore IL-2 production from these T cells was significantly low for every antibody. Nevertheless, the combinational use of anti-hPD-1 and anti-hPD-L1 at each EC₅₀ showed greater enhancement of IL-2 production than a single use of each antibody at its own EC₉₀.'

13. The PD-L1 in Figure 5e should be substituted with PD-L2.

As pointed out, we correct the legend.

14. For consistency I would like to see the % T cell forming PD-1 microclusters upon inhibitors application based on the images shown in Supplementary Fig. 4a.

To follow the reviewer's recommendations, we depicted new bar graphs as Supplementary Fig. 4b.

15. The authors should provide information how the increased ratio of IL-2 in Fig. 7b and statistical analysis of the data presented in Fig. 7c.

To answer the reviewer's comments, we revised the statistical analysis in Fig. 7b and added the statistical analysis in Fig. 7c (new Fig. 7f).

16. There is a complete lack of information regarding the formation of SLB a central piece of the imaging system. It will be of interest to know the exact procedure, the type of substrate used as well as the possibility of using high-throughput formats.

To follow the reviewer's recommendations, we added the information of the imaging system to the Materials and Methods.

17. In general, throughout the manual there is a lack of citations.

We apologize for lacking the papers to be referred. We added 20 papers as number 2, 3, 4, 5, 6, 9, 10, 11, 12, 15, 16, 17, 18, 19, 20, 21, 28, 29, 32 and 33 in the reference.

Revising the manuscript stated above, the original figure numbers are changed from Fig. 2f to 2g, from Fig. 4b to 4c, from Fig. 4c to 4e, from Fig. 4d to 4f, from Fig. 4e to 4g, from Fig. 4f to 4h, from Fig. 4g to 4i, from Fig. 7c to 7f, from Fig. 7d to 7g, from Fig. 7e to 7h, from Supplementary Fig. 1b to Supplementary Fig. 1f, from Supplementary Fig. 1c to Supplementary Fig. 1g, from Supplementary Fig. 4b to Supplementary Fig. 4c, from Supplementary Fig. 5a to Supplementary Fig. 5c, from Supplementary Fig. 5b to Supplementary Fig. 5g, from Supplementary Fig. 6a to Supplementary Fig. 7a, from Supplementary Fig. 6b to Supplementary Fig. 7c, from Supplementary Fig. 6c to Supplementary Fig. 7d, from Supplementary Fig. 6d to Supplementary Fig. 7e and from Supplementary Table 1 to Supplementary Table 3.

The figure number have been changed as summarized in below table.

before	after
	2f
2f	2g
	4b
4b	4c
	4d
4c	4e
4d	4f
4e	4g
4f	4h
4g	4i
	7c
	7d
	7e
7c	7f
7d	7g
7e	7h
	Sup. 1b

	Sup. 1c
	Sup. 1d
	Sup. 1e
Sup. 1b	Sup. 1f
Sup. 1c	Sup. 1g
	Sup. 1h
	Sup. 3c
	Sup. 4b
Sup. 4b	Sup. 4c
	Sup. 4d
	Sup. 4e
	Sup. 4f
	Sup. 4g
	Sup. 4h
	Sup. 4i
	Sup. 4j
	Sup. 5a
	Sup. 5b
Sup. 5a	Sup. 5c
	Sup. 5d
	Sup. 5e
	Sup. 5f
Sup. 5b	Sup. 5g
	Sup. 6a
	Sup. 6b
Sup. 6a	Sup. 7a
	Sup. 7b
Sup. 6b	Sup. 7c
Sup. 6c	Sup. 7d
Sup. 6d	Sup. 7e
	Sup. Table 1
	Sup. Table 2
Sup. Table 1	Sup. Table 3

REVIEWERS' COMMENTS

Reviewer #1 (Remarks to the Author):

The authors have addressed the concerns raised by the reviewers with additional experiments.

Reviewer #2 (Remarks to the Author):

The authors have improved the manuscript according to the suggestions from all reviewers. Most of my questions were addressed and further explanation was provided.

Although further improvements of the image analysis will improve the study, this is valuable work that would be useful for the community and might initiate the development of alternative combinatorial drug discovery screening platforms.

The following are the point-by-point response to the reviewers' comments.

Reviewer #1 (Remarks to the Author):

The authors have addressed the concerns raised by the reviewers with additional experiments.

We thank the reviewer for reviewing our manuscript.

Reviewer #2 (Remarks to the Author):

The authors have improved the manuscript according to the suggestions from all reviewers. Most of my questions were addressed and further explanation was provided.

Although further improvements of the image analysis will improve the study, this is valuable work that would be useful for the community and might initiate the development of alternative combinatorial drug discovery screening platforms.

We thank the reviewer for reviewing our manuscript.